# Reference-based chemical-genetic interaction profiling to elucidate small molecule mechanism of action in *Mycobacterium tuberculosis*

We previously reported an antibiotic discovery screening platform that identifies whole-cell active compounds with high sensitivity while simultaneously providing mechanistic insight, necessary for hit prioritization. Named PROSPECT, (PRimary screening Of Strains to Prioritize Expanded Chemistry and Targets), this platform measures chemical-genetic interactions between small molecules and pooled *Mycobacterium tuberculosis* mutants, each depleted of a different essential protein. Here, we introduce Perturbagen CLass (PCL) analysis, a computational method that infers a compound's mechanism-of-action (MOA) by comparing its chemical-genetic interaction profile to those of a curated reference set of 437 known molecules. In leave-one-out cross-validation, we correctly predict MOA with 70% sensitivity and 75% precision, and achieve comparable results (69% sensitivity, 87% precision) with a test set of 75 antitubercular compounds with known MOA previously reported by GlaxoSmithKline (GSK). From 98 additional GSK antitubercular compounds with unknown MOA, we predict 60 to act via a reference MOA and functionally validate 29 compounds predicted to target respiration. Finally, from a set of ~5,000 compounds from larger unbiased libraries, we identify a novel QcrB-targeting scaffold that initially lacked wild-type activity, experimentally confirming this prediction while chemically optimizing this scaffold. PCL analysis of PROSPECT data enables rapid MOA assignment and hit prioritization, streamlining antimicrobial discovery.

For over half a century, standard chemotherapy for tuberculosis (TB) has been a four drug, six-month regimen[1]. The cornerstones of this regimen have been the RNA polymerase inhibitor rifampin (RIF) and the cell-wall targeting prodrug isoniazid (INH). Resistance to these drugs is on the rise, as well as increasing resistance to second-line therapies used in multidrug-resistant (MDR) cases. Recently, a few new antitubercular drugs have been approved by the FDA, leading to the introduction of a more effective regimen for the treatment of MDR-TB that includes the ATP synthase inhibitor bedaquiline along with

pretomanid and linezolid[2]. Nevertheless, there continues to be a need for novel drugs, particularly with new mechanisms of action to circumvent existing resistance mechanisms.

Many conventional antibiotic discovery efforts have historically relied on biochemical assays to identify molecules against specific targets of interest, but such compounds are typically inactive against whole cells[3]. On the other hand, the alternative approach of whole-cell screening for compounds that kill live cells is also challenging. One issue is the limited sensitivity of such screens, as even potentially

e-mail: jgomez@broadinstitute.org; hung@molbio.mgh.harvard.edu

promising compounds often have very low potency before optimization. The most significant limitation, however, is that molecules are prioritized based on potency and chemical properties, devoid of any biological insight that would enable prioritization based on mechanisms of action (MOAs) of known interest. In the case of *Mycobacterium tuberculosis* (*Mtb*), identified small molecule candidates have thus been biased towards a small number of MOAs[4,5], despite the fact that *Mtb* has approximately 600 essential genes[6,7] representing a large diversity of biological processes. Without early MOA information, not only are subsequent extensive chemistry campaigns more challenging because of the lack of insight from structural target engagement, but they also often result in frustration when much later target identification reveals an MOA of little interest. Thus, early target elucidation to inform candidate selection and prioritization would be transformative.

We have previously reported on a novel systems chemical biology strategy for the identification of small molecule candidates with antitubercular activity that solves three problems at once: (1) it is more sensitive and generates 10-fold more hits compared to conventional methods screening wild-type bacteria[8], (2) it finds hits that target a variety of protein targets, potentially any of the 600 *Mtb* essential proteins, and, importantly, (3) it provides MOA insight for compounds, allowing prioritization of hits before further costly development and target validation[8]. This method, PROSPECT (PRimary screening Of Strains to Prioritize Expanded Chemistry and Targets), couples small molecule discovery to MOA information by screening each small molecule against a pool of hypomorphic *Mtb* strains, each engineered to be proteolytically depleted of a different essential protein[9]. The degree to which the growth of each hypomorph in the pool is affected by a compound is measured using next-generation sequencing to quantify the change in abundances of hypomorph-specific DNA barcodes. The impact of a particular chemical perturbation on a genetically engineered hypomorphic strain manifests as a chemical-genetic interaction (CGI). The readout for each compound-dose condition (i.e., screened compound at a specific concentration) is thus a vector of the responses of the collection of hypomorphs, i.e., a vector of CGIs, which we call a CGI profile.

Not only does the potential hypersensitivity of hypomorphic strains enable the discovery of active small molecules that would elude wild-type screening, but the identity of the hypomorphs that are sensitive to a small molecule can also provide information about its MOA. In principle, in a hypomorph where the level of some essential protein is sufficiently low, any further reduction of the protein level, even a small one, can lead to cell death. This suggests that a hypomorph for a given essential gene would be specifically vulnerable to compounds that target and inhibit the corresponding gene product directly, its pathway, or adjacent pathways that interact with it[10–15]. We previously demonstrated the ability of PROSPECT to identify scaffolds to new targets with the discovery of a bactericidal pyrimidyl-cyclopropane-carboxamide inhibitor of the essential EfpA transporter[8], with the EfpA hypomorphic strain uniquely sensitized. However, because of the complex genetic interactions within a cell, it is rare to be able to identify the target directly based only on a single, most sensitized hypomorphic strain.

There are two ways to address the challenge of inferring the MOA of compounds from their CGI profiles. In a reference-based approach, the CGI profiles can serve as fingerprints of chemical perturbations without the need to understand the biology encoded in specific strain identities; inference about an unknown compound is based on the similarity of a CGI profile to the CGI profiles of one or more reference compounds whose target/MOA is already known. Alternatively, in a more daunting reference-free manner, one can try to infer the MOA based on the biology encoded in the entire CGI profiles of a compound, incorporating in some way the interactions and relationships between genes and pathways. While a reference-free approach is hindered by limited understanding of all biological interactions within the

cell, a reference-based approach is constrained by the limited availability of known compounds with annotated MOAs. The significant value of a reference-based approach nevertheless lies in its ability to rapidly identify (1) new scaffolds for validated, valuable targets that can circumvent existing resistance, (2) scaffolds that work by known MOAs of low interest thereby enabling their early deprioritization, and, by the process of elimination, (3) scaffolds that work by completely novel MOAs that are not represented in the reference set.

Here we report a reference-based approach for MOA prediction, termed Perturbagen CLass (PCL) analysis. We curated a reference set of 437 compounds with published, annotated MOA and known or possible anti-tubercular activity. We applied PROSPECT to obtain CGI profiles of all compounds in this reference set in dose-response. The reference set data were used to develop and optimize PCL analysis, the performance of which was evaluated using a leave-one-out analysis (70% sensitivity, 75% precision). We then applied PROSPECT screening in dose-response and PCL analysis to a collection of 173 compounds previously reported by GlaxoSmithKline (GSK) to have potent anti-tubercular activity[16,17]. The fraction of compounds within this collection whose MOAs have been annotated since their original publication served as a test set to further evaluate the performance of the PCL method (69% sensitivity and 87% precision). Meanwhile, on the unannotated part of the GSK set, PCL analysis newly assigned putative MOAs to 60 compounds from 10 MOA classes. Across the entire GSK set, a remarkably large fraction (38%; 65 compounds) were high-confidence PCL matches to known inhibitors of QcrB, a subunit of the cytochrome *bcc-aa3* complex involved in respiration, including both well-validated scaffolds as well as structurally novel inhibitors. We validated the predicted QcrB MOA of the majority of these by confirming their loss of activity against mutants carrying a *qcrB* allele known to confer resistance to known QcrB inhibitors and their increased activity against a mutant lacking cytochrome *bd*, hallmarks of QcrB inhibitors[18]. Finally, we applied PROSPECT screening in dose-response and PCL analysis to a set of over 5000 compounds we had previously identified based on potency or strain specificity from single-dose, PROSPECT screens of unbiased chemical libraries that had not been preselected for antitubercular activity. We followed up on one of these compounds, a novel pyrazolopyrimidine scaffold with no significant wild-type activity in the screen, but with a high confidence PCL-based prediction to target the cytochrome *bcc-aa3* complex. We confirmed that the QcrB subunit of the complex is indeed the target and achieved potent wild-type activity through chemistry efforts. Taken together, PCL analysis can predict MOA both for pre-identified, potent antitubercular compounds as well as for molecules from screening of unbiased libraries that result in the identification of candidates initially lacking wild-type activity—but that can be subsequently achieved through chemical synthesis. This approach constitutes an efficient, high-throughput strategy for yielding new, potent antitubercular compounds with annotated MOA.

## Results
### Curation of a reference set with annotated MOA
To learn how to best interpret the complex PROSPECT output data, we extensively mined the published literature to assemble a reference set of 437 compounds with annotated MOAs and known or predicted anti-tubercular activity (Fig. 1a, Supplementary Data 1). We included anti-tubercular active molecules with MOAs with varying degrees of evidence, from strong mechanistic validation to in silico protein docking. This included established antitubercular compounds (e.g., isoniazid, ethambutol, bedaquiline) as well as advanced (e.g., Q203 and SQ109) and less-developed lead compounds (e.g., numerous MmpL3 and DprE1 inhibitors, BRD4592 targeting TrpAB[19], and benzofuran Pks13 inhibitors[20]). We also included well-characterized antimicrobials with broad-spectrum activities including anti-tubercular activity (e.g., fluoroquinolones, macrolides, beta-lactams), and some with no or

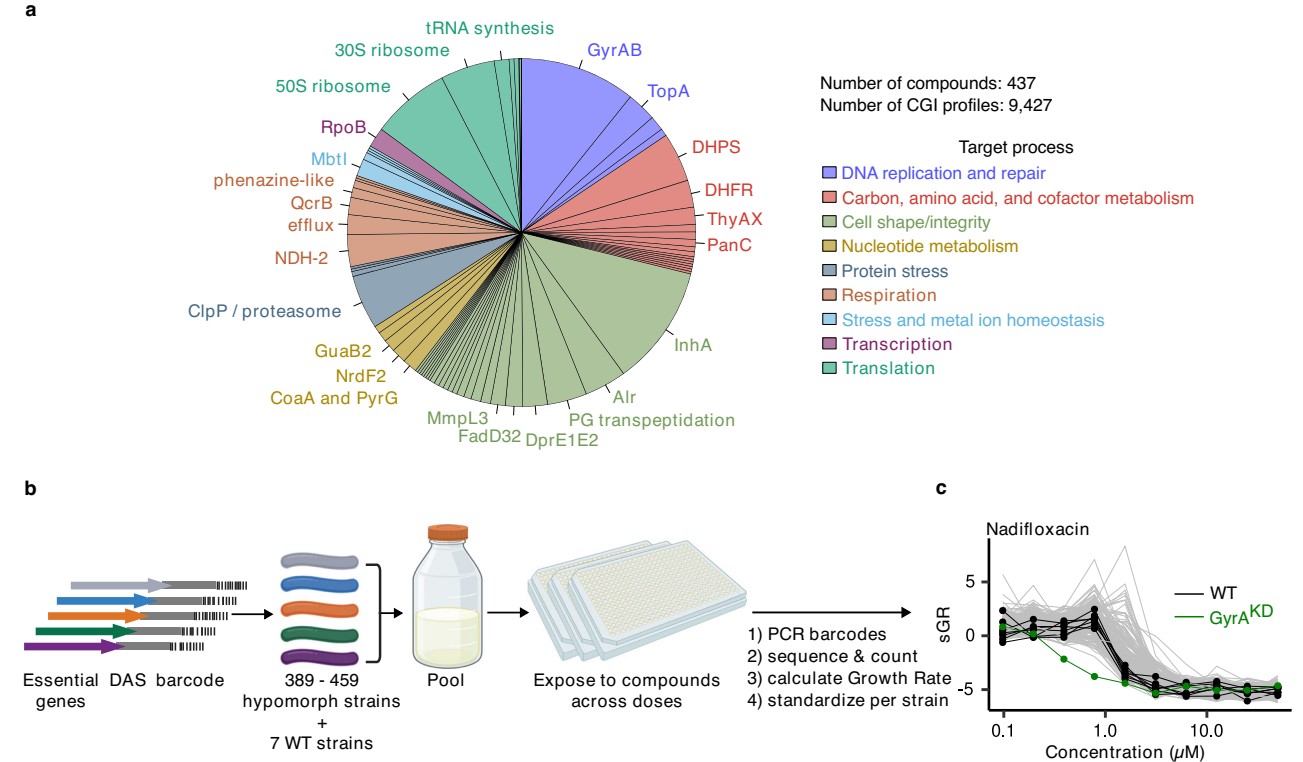

**Fig. 1 | PROSPECT screen on reference collection of compounds with known antibacterial activity and annotated MOA. a** Breakdown of the reference set, which comprises 437 compounds from 71 annotated mechanisms of action (MOAs) that target 9 high-level processes. **b** Schematic of the PROSPECT (PRimary screening Of Strains to Prioritize Expanded Chemistry and Targets) pipeline (Created in BioRender. Lab, H. (2025) https://BioRender.com/oa7xnfj). Barcoded hypomorph strains, depleted for one of *Mtb*'s essential proteins, are pooled, incubated for 14 days in 384-well plates containing the compound library, and then heat-killed and lysed. Barcodes are PCR amplified and amplicons sequenced to yield barcode counts as a measure of strain census in each well. Compound-induced Growth Rate (GR) for each strain in response to each condition is based on the ratio of condition- and vehicle- numbers of doublings. Strain distributions are then quantile normalized and robust z-scored across all conditions to calculate standardized growth rate (sGR). **c** Example standardized growth rate (sGR) chemical-genetic interaction (CGI) profiles across different concentrations of the fluoroquinolone nadifloxacin for 7 wild-type (WT) H37Rv barcoded strains (black), GyrA hypomorph (green), and 332 other hypomorphs in the pool (gray). Source data are provided as a Source Data file.

limited activity reported for wild-type *Mtb*. We also included compounds that have biochemical evidence for an antibacterial target without documentation of whole-cell, on-target activity in case activity in hypomorphs might be observed (e.g., alloxydim and haloxyfop herbicides putatively targeting AccD6)[21,22]. Finally, we included some compounds that have MOAs validated in eukaryotic cells but are also known or anticipated to have antimicrobial activity, particularly at higher concentrations (e.g., camptothecin which targets human topoisomerase I)[23,24]. In total, the reference set involved molecules spanning 71 distinct MOAs, with the important caveats that molecules with less well validated MOAs might be inaccurately or incompletely annotated, or molecules with MOAs described in some species could work by a different MOA in *Mtb*.

We performed several waves of PROSPECT screening as previously described in ref. 8 across a 10-point, 2-fold dilution series, with each specific compound-dose referred to as a condition (Fig. 1b). Briefly, each compound-dose condition is applied to a pool of hypomorphic *Mtb* strains, each uniquely barcoded and proteolytically depleted of one essential protein. Strains are allowed to grow for 14 days, when barcode DNA is PCR amplified and sequenced and barcode reads are counted to estimate occurrence of each strain in response to compounds. Since the initial report describing PROSPECT using pools of 100–150 barcoded, genetically engineered hypomorphs, we have expanded the screening pool sizes to include between 389 and 459 hypomorphic strains per screen, representing approximately 75% of the genes necessary for *Mtb* to grow in vitro[6,7]

(Fig. 1b, Supplementary Data 2). Additionally, seven uniquely barcoded but otherwise unmodified H37Rv (wild-type) strains were included in the pool as controls. After excluding strains that were not present in all screening waves or that grew unreliably, we used 340 strains (333 hypomorphs and 7 wild-type controls) for downstream analysis (Supplementary Note 2).

## Growth rate metrics
Previously[8] we quantified compound effect on strains using a log2-fold change (L2FC) metric, based on the ratio of barcode counts corresponding to each strain in a compound-dose condition to the counts for that strain in the vehicle control (Supplementary Fig. 1a, Supplementary Note 1). This corresponds to the condition-induced change in the number of doublings of a strain during the 14-day assay period (rifampin positive control estimates count at time zero). This approach, however, presents a problem when comparing strains that have different baseline (vehicle-control) growth rates[25] as faster-growing strains exhibit larger absolute change in the number of doublings for the same relative effect (Supplementary Figs. 1d, 3a). Indeed, the baseline growth rates of different hypomorphs spanned a range of doubling rates, from 1.8 (RfbE) to 5.4 (Acn) doublings in 14 days, with differences attributed to varying degrees of protein knockdown and variable dependencies of growth on the level of each essential protein[6] (Supplementary Fig. 2, Supplementary Data 2).

To address this problem, we implemented a metric of (dose-dependent) compound-induced growth rate, GR, which measures

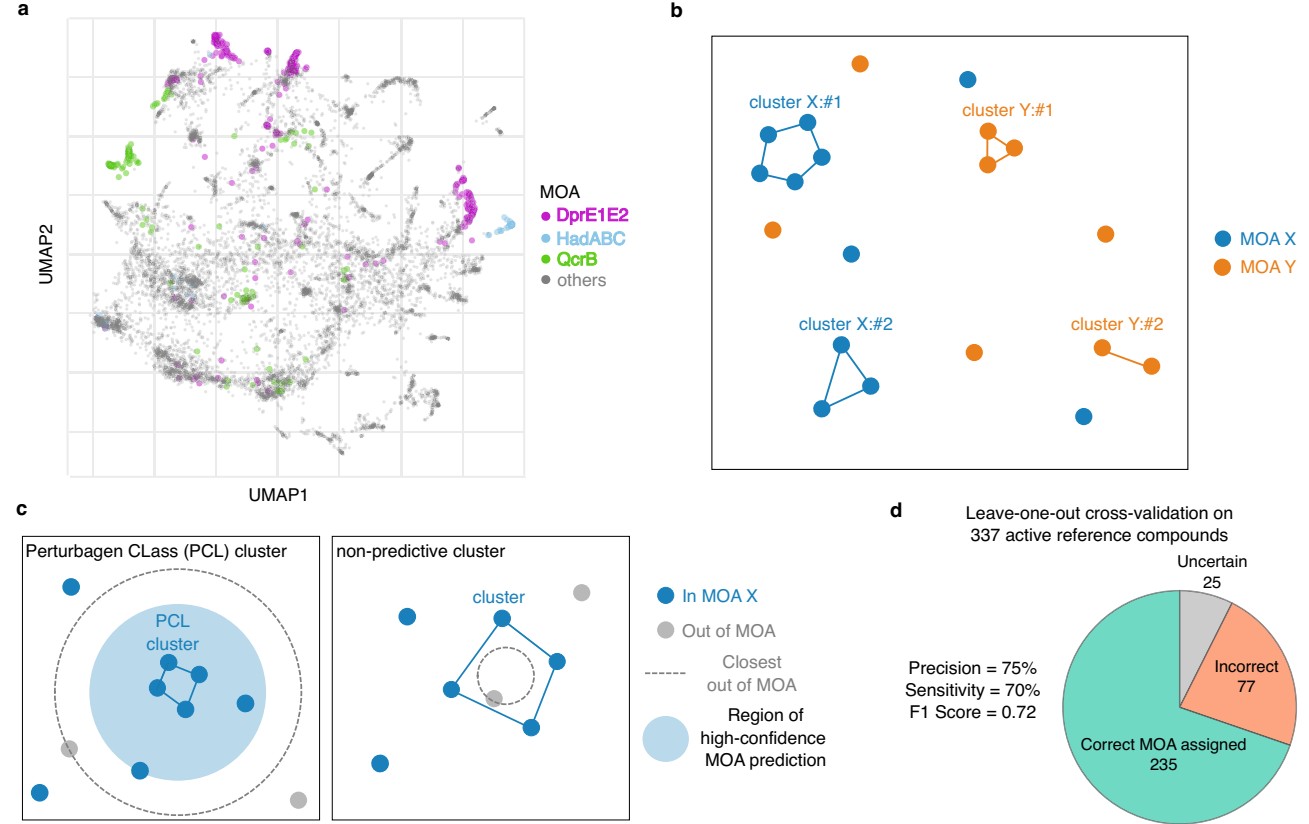

**Fig. 2 | Perturbagen CLass (PCL) analysis for reference-based mechanism of action (MOA) prediction. a** Visualization using Uniform Manifold Approximation and Projection (UMAP) of all reference set compound-dose chemical-genetic interaction (CGI) profiles reveals MOA-based clustering of compound-dose conditions. Three exemplary MOAs are highlighted: DprE1-DprE2 complex (purple), HadABC (light blue), and QcrB (green). Grey circles represent reference CGI profiles from all other MOAs. The UMAP representation of the data is shown here for illustration purposes only; none of the steps in the PCL analysis method depend on this representation. **b** Schematic of the results of spectral clustering of each MOA category. Circles represent CGI profiles from two MOAs, X (blue) and Y (orange), each yielding two clusters (connected circles) and some singleton CGI profiles. **c** (left) Schematic of a high-confidence prediction region (light blue shaded circle). The blue circles all share an MOA X, which is different from any MOA represented

by the light gray circles. The blue circles connected by lines mark the cluster for which the high-confidence region is drawn. The radius of the high-confidence region for a cluster is defined as the largest distance (lowest similarity) between a CGI profile and the cluster such that all profiles contained within that radius belong to the same MOA X. Clusters for which such a high-confidence region exists are called Perturbagen CLass (PCL) clusters. Similarity score between a given CGI profile and a cluster is defined as the median of the correlations between the CGI profile and all the cluster profiles. (right) Schematic example of a non-predictive cluster to which the most similar CGI profile is out-of-MOA. Such clusters are considered not reliable for MOA prediction and are discarded. **d** Performance statistics of PCL analysis method for MOA prediction on 337 active reference compounds in leave-one-out cross-validation (LOOCV). Source data are provided as a Source Data file.

growth rate relative to vehicle control[26]. Unlike L2FC, which is the difference in the number of doublings between condition and vehicle, GR is a function of their ratio: $GR = 2^{\frac{\text{number of doublings in condition}}{\text{number of doublings in vehicle}}} - 1$ (Supplementary Note 2). It therefore projects the effect of each condition onto a scale from 0 to 1, where 1 corresponds to uninhibited growth (vehicle control) and 0 corresponds to 100% inhibition (rifampin control), to normalize the dynamic range of strain growth rates (Supplementary Fig. 1b). Of note, due to the stability of the DNA barcode used to quantify census (i.e., barcodes from dead cells are also counted), PROSPECT cannot differentiate bacteriostatic (GR = 0) and bactericidal effects (theoretical GR < 0). By using the ratio of doubling numbers, GR allows us to compare the effects of a compound-dose condition on different hypomorphs irrespective of their baseline growth rate (Supplementary Figs. 1e, 3b).

Further, to reveal the magnitude of the inhibitory effect of a given compound-dose condition on a strain relative to the effect of all other conditions on that strain across the entire dataset, we first quantile normalized[27] the distributions of GR scores across strains and then standardized (z-scored) them for each strain across all conditions, resulting in sGR (standardized GR, Fig. 1c, Supplementary Figs. 1c, f–g, 3c, d, Supplementary Note 3). The vector of sGRs for all strains in the

pool in response to a given condition (compound at a given concentration) makes up a chemical-genetic interaction (CGI) profile, used in all subsequent analyses[28].

## PCL analysis: construction of PCLs

The basic premise behind a reference-based approach to MOA assignment from PROSPECT data is that compounds that share a common MOA generally elicit similar and specific patterns of strain sensitization, as reflected in their CGI profiles[8,29]. This is illustrated in a two-dimensional uniform manifold approximation and projection (UMAP) of all CGI profiles in the reference set, where three exemplary MOAs highlighted in color demonstrate expected MOA-specific aggregation (Fig. 2a, Supplementary Note 3). A simple reference-based strategy might assign MOA to a query compound by identifying the reference CGI profile most highly correlated to any of the query CGI profiles (i.e., 1-nearest neighbor). However, such a strategy fails to acknowledge that not all compound-dose conditions have equally informative CGI profiles, as seen by the scattering of some CGI profiles of the highlighted MOAs (Fig. 2a). Specifically, there are extremely active or completely inactive doses of reference compounds, wherein no MOA-specific strain sensitization can be observed. To counter this,

our method, PCL analysis, relies on corroboration: MOA inference is made based on similarity of a query CGI profile to groups of at least two similar CGI profiles of the same MOA.

To systematically identify all such groups, we applied spectral clustering[30–32] to the CGI profiles in each of the 71 MOA categories in the reference set (Fig. 2b). Here similarity of profiles was defined by the Pearson correlation between them (Supplementary Fig. 4, Supplementary Note 7). Often, CGI profiles generated from compounds with the same MOA fall into several clusters, separating, for example, different levels of compound activity (Supplementary Fig. 4b). Furthermore, some of the conditions do not cluster with any other condition - they form singletons. We consider their CGI profiles to be less reliable since they are not corroborated by any other compound-dose condition. Indeed, such singletons are often conditions that are either inactive or that fully inhibit the entire pool (Supplementary Fig. 5, Supplementary Note 4). For this reason, in the next step we considered only clusters containing at least two CGI profiles.

Using spectral clustering, we generated 1947 clusters of two or more CGI profiles that are similar to one another and are associated with a particular MOA. Our reference-based approach aims to infer the MOA of an unknown compound by examining its CGI profiles (at different doses) and mapping it to its most similar cluster. To that end, we defined the similarity score between a given CGI profile and a cluster to be the median of the correlations between the CGI profile and all the profiles within the cluster (Supplementary Note 8). It is intuitively convenient to think about similarity in terms of distance, such that profiles that are more similar to one another are closer. Using that language, we asked what is the radius of the largest MOA-pure region around a cluster, i.e., which contains only CGI profiles from the cluster's MOA. We then defined the corresponding similarity score to be the high-confidence threshold for that cluster (Fig. 2c left, Supplementary Note 9), wherein if a CGI profile in question is at least as similar to a cluster as that threshold, the MOA of the cluster is assigned with high-confidence (Fig. 2c left). For 807 of the clusters, the closest CGI profile was from a different MOA, and therefore these clusters could not be assigned a high confidence (MOA-pure) region and were discarded (Fig. 2c right). Similarly to singletons, we found that these non-predictive clusters were typically composed of inactive doses or doses that kill the entire pool and did not use them for further analysis (Supplementary Fig. 5). After discarding the non-predictive clusters, 1140 clusters remained to which we could assign a non-zero high-confidence threshold. We named these Perturbagen CLass (PCL) clusters following Subramanian et al.[33] These 1140 PCLs represented 68 MOAs out of the initial 71 MOAs in the reference set, and the number of CGI profiles in each PCL varied from 2 to 51 with a median of 3.5 (Supplementary Fig. 6, Supplementary Data 3).

## PCL analysis: MOA prediction and cross validation

Having constructed PCL clusters, we then sought to predict MOAs for unknown compounds based on PCLs. We assigned a high-confidence MOA prediction to a compound if one of its doses had a similarity score to one of the PCLs which was above the high-confidence threshold. The MOA of compounds with only below-confidence-threshold predictions were labeled uncertain. We further defined PCL confidence scores to estimate how likely it is that a test compound shares an MOA with a given PCL. PCL confidence score is defined as the fraction of in-MOA compounds among all the reference compounds that were at least as similar to the PCL as the test compound, with scores ranging between 0 and 1 and a confidence score of 1 corresponding to a high-confidence MOA assignment (Supplementary Figs. 7, 8, 9, Supplementary Notes 9, 10, 11).

To estimate the performance of this MOA-prediction method, we applied a leave-one-out cross-validation (LOOCV)[34] approach, withholding each of the reference compounds individually when forming PCLs and from the process of defining the confidence regions, and

then predicting the MOA of the withheld compound (Supplementary Note 12). We then assessed the accuracy of the predictions. We excluded from this analysis: (1) 19 MOAs that were only represented by one compound as these cannot be validated in any cross-validation scheme, and (2) compounds with little to no activity as their CGI profiles lack informative data for meaningful MOA predictions. We therefore performed LOOCV analysis on the 337 reference compounds that were active (defined as at least one strain having $GR \le 0.3$ at the highest tested dose) (Supplementary Note 6). Out of these 337 reference compounds, we assigned the correct MOA in LOOCV to 235 compounds, an incorrect MOA to 77 compounds, and 25 compounds were uncertain (sensitivity = 70%, precision = 75%, F1 score = 0.72, Fig. 2d, Supplementary Table 1, Supplementary Data 4). On the other hand, for the reference compounds that were inactive, as expected, MOA prediction performance was poorer (sensitivity = 34%, precision = 26%, F1 score = 0.29, Supplementary Table 1, Supplementary Data 4, Supplementary Fig. 10).

Of note, without the corroboration provided by spectral clustering of multiple reference CGI profiles into PCLs, applying the aforementioned 1-nearest neighbor approach to the 337 active reference compounds could only assign the correct MOA in LOOCV to 215 compounds and an incorrect MOA to 122 compounds (sensitivity = 64%, precision = 64%, F1 score = 0.64, Supplementary Table 1, Supplementary Note 12). Furthermore, in the absence of clustering of MOAs to PCLs as a critical step, i.e., naively grouping all CGI profiles from each MOA into one large PCL cluster, MOA prediction performed significantly worse over the active reference compounds due to median similarity scores being diluted by inactive compound-dose conditions (sensitivity = 11%, precision = 69%, F1 score = 0.19, Supplementary Table 1, Supplementary Note 12).

## PCL analysis on annotated GSK compounds

We applied PROSPECT screening in a 10-point dose-response and then PCL analysis to a collection of 173 antitubercular compounds, provided by GlaxoSmithKline (GSK), comprising of whole-cell active, non-cytotoxic compounds identified in two previously published screens[16,17] (Supplementary Data 5). [35–42] Of note, the blinded set of 173 compounds provided by GSK included 8 compounds that overlapped with our reference collection. These included an MmpL3 inhibitor[43], an MmpL3/EchA6 inhibitor[44,45], 4 PanK and PyrG inhibitors[36,37], a DHPS inhibitor sulfaphenazole[46], and a DHFR inhibitor[40,47]. Since the public releases of these compound sets, they have been characterized by a variety of methods both at the level of individual scaffolds as well as in large scale analyses, including metabolomic profiling[35], biochemical assays[36,37], screening of overexpression strains[38,39], chemogenomic predictions[40], or in infection models[41,42], resulting in the publication of MOAs (with varying degrees of support) for 75 of the 173 GSK compounds. We used these 75 annotated compounds as a held-out test set to further benchmark PCL analysis performance (Fig. 3a, Supplementary Data 5).

We made high-confidence predictions for 60 (80%) of these 75 characterized compounds, assigning an MOA consistent with the published MOA for 52 of them (69% sensitivity, 87% precision) (Fig. 3a, b). Among the compounds correctly assigned were published folate inhibitors including several diaminotriazines[47] and a sulfaphenazole. We also correctly identified a gyrase inhibitor[48], DprE1 inhibitor[38], and a known thiazole pyridine PyrG/PanK inhibitor. Importantly, although the folate and PyrG/PanK inhibitors are structurally related to compounds in the reference set, these MOA predictions were made blinded to chemical structure.

MmpL3, a mycolic acid transporter, is the target of several recently reported inhibitors with a broad range of pharmacophores[49]. These inhibitors possess properties that appear to vary based on the scaffold, including the ability to kill non-replicating *Mtb* and dissipate the proton motive force[39,49,50], and some have been shown to have

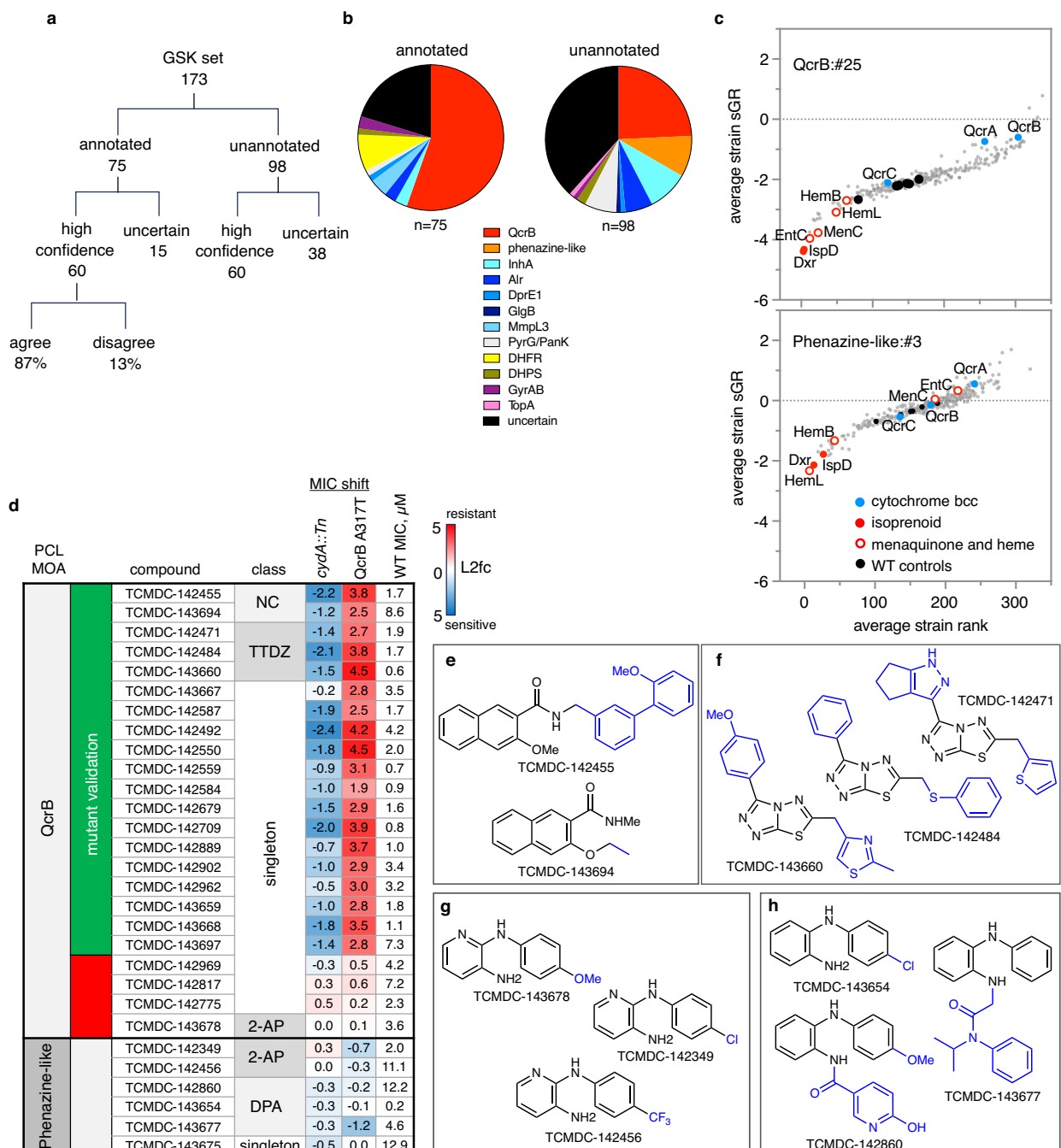

**Fig. 3 | Mechanism of action (MOA) assignments from the GSK set.**
**a** Perturbagen Class (PCL) MOA predictions for GlaxoSmithKline (GSK) set.
**b** Distribution of high-confidence PCL MOA assignments. **c** MOA-specific sensitization of strains in PCL clusters from respiration inhibitors. y-axis = mean strain standardized growth rate (sGR) across all conditions in the PCL cluster, x-axis = average strain rank. QcrB:#25 has 29 chemical-genetic interaction (CGI) profiles from 4 unique compounds; Phenazine-like:#3 has 24 CGI profiles from 4 unique compounds. Cytochrome *bcc* components are highlighted in blue, isoprenoid synthetic enzymes in solid red, open red circles show additional strains linked to menaquinone and heme synthesis, solid black circles are wild-type controls, and the rest of the hypomorphs are gray circles. **d** Confirmation of activity of predicted novel QcrB inhibitors. Heatmap of minimum inhibitory concentration (MIC) shifts

against *cydA*::Tn and QcrB A317T mutants relative to wild-type (WT) H37Rv. PCL predictions shown in column 1. In column 2, compounds with MIC-shift-based confirmation of QcrB MOA in green; red indicates MIC shifts do not support QcrB as the target. Column 4 shows the scaffold class: napthalene carboximides (NC); triazolothiadiazoles (TTDZ); diphenylamines (DPA); anilopyridines (2-AP). In columns 5-7, the log2-fold change (L2fc) in MIC for the two mutants relative to the wild-type MIC is shown with blue showing sensitization and red showing resistance, along with the wild-type MIC. MIC values are curve fits to average of 2 replicates. **e–h** Scaffolds of clusters of novel respiration inhibitors. **e** NC inhibitors of QcrB. **f** TTDZ inhibitors of QcrB. **g** 2-APs displaying phenazine-like activity. **h** DPAs displaying phenazine-like activity. Source data are provided as a Source Data file.

secondary targets[45,51–53]. We made high-confidence predictions for 7 of 11 published MmpL3 inhibitors, but correctly identified only 3; interestingly, 3 of the 4 incorrect calls were high-confidence predictions to a different cell-wall MOA, alanine racemase (Alr) inhibition, which

might suggest a genetic interaction between MmpL3 and Alr. Alternatively, many of the Alr-annotated inhibitors in the reference set were annotated as such based on biochemical evidence[54–56]; in the context of whole cells, these Alr reference compounds may have additional

targets which could possibly include MmpL3[54,56]. Taken together, these discordant calls illuminate potential challenges in assigning cell wall MOAs, underscoring the interdependence of cell-wall synthetic processes. Alternatively, misannotation of compounds in the reference set could likewise lead to discordance. Thus, the quality of the reference set annotation can pose challenges particularly when annotations in the reference set are weak at best, incorrect at worst.

QcrB has also been a frequently identified target of potent, non-toxic molecules identified through whole-cell screening efforts. Thus, perhaps unsurprisingly, 42 of the 75 now annotated and published compounds have been validated as QcrB-targeting scaffolds (see Supplementary Data 5), including imidazopyridines[44,57–59]; quinolinyloxyacetamides[60,61]; piperazines[62]; triazolopyrimidines[63]; quinazolines[64]; and one phenoxyalkylbenzimidazole[65]. PROSPECT and PCL analysis made accurate high-confidence QcrB predictions for 39 (93%) of these. Despite having only two of these six scaffolds in the reference set, we were able to correctly assign MOA to compounds of the remaining four chemical classes, thus demonstrating the power of PCL analysis to accurately predict MOA directly from PROSPECT screening data.

## PCL analysis on unannotated GSK compounds
Even a decade after their initial publication, most of the GSK compounds (the remaining 98 out of the 173) have not yet had a confident MOA assignment in the published literature. Application of PCL analysis to these compounds thus has the potential to annotate the unexplored target space of this important collection. PROSPECT screening and PCL analysis resulted in high-confidence MOA calls for 60 (61%) of these 98 compounds (Fig. 3a, b). Compared to the published, annotated compounds, the smaller fraction of high-confidence predictions for these unannotated compounds (61% vs 80%) likely reflects their potential enrichment for compounds with novel MOAs that are not represented in the reference set. Thus, an important corollary to PCL analysis is that the absence of high-confidence predictions can be used to identify molecules with potentially novel activities.

Among the 60 new, high-confidence predictions made, 10 unique MOAs were represented. Most of these new PCL-based MOA predictions fall within three major processes: cell wall synthesis, DNA replication/nucleotide metabolism, and respiration (Fig. 3b, Supplementary Fig. 11a). The cell-wall predictions included new, putative, InhA inhibitor scaffolds, novel Alr inhibitors (with the aforementioned caveat regarding the uncertainty of MOA annotation of Alr reference compounds), along with a DprE1/DprE2 inhibitor and a GlgB inhibitor. The DNA replication/nucleotide metabolism-related compounds included thiazoles which are predicted to share a target with the GSK set-derived reference thiazoles previously reported to target PyrG and PanK. Additional compounds predicted to target DNA replication/nucleotide metabolism include one PyrG/PanK targeting sulfonamide, an isoquinoline gyrase inhibitor, and a TopA inhibitor.

Respiration MOAs were the most frequently predicted among the 60 high-confidence predictions, with 33 structurally diverse compounds predicted to target either QcrB (24) or displaying phenazine-like activity (9). Inhibiting the critical process of respiration has been of growing interest[66]. Of note, the respiration inhibitors identified in this analysis underscored the critical principle underlying PCL analysis: MOA assignment is based on correlation with a canonical CGI profile pattern and does not require the sensitization of the direct target of a compound. In the case of QcrB inhibitors, the assignment is based on a strong interaction between isoprenoid biosynthesis and sensitivity to respiratory inhibition, with marked sensitization of strains hypomorphic for Dxr and IspD, two essential enzymes[67] that execute the first two committed steps in the isoprenoid biosynthetic pathway[68] (Fig. 3c). Isoprenoids are required for the synthesis of menaquinone

and heme $a$[69], two molecules critical to the electron transport chain (ETC)[70]. In addition, the MenC hypomorph, also impaired in menaquinone synthesis[71], is highly sensitized, along with EntC, which plays a role in both menaquinone synthesis and iron acquisition. Enzymes in heme synthesis (HemB and HemL) are also sensitized. Together, this suggests that bottlenecks in the de novo synthesis of menaquinone and heme, two central molecules in the electron transport chain, result in sensitization to QcrB inhibitors. Notably, hypomorphs depleted for components of cytochrome $bcc$ were not among the strains that displayed hypersensitivity to known QcrB inhibitors (Fig. 3c). In fact, the QcrA and QcrB hypomorphs ranked among the 15% least sensitized strains in a large PCL defined by known QcrB inhibitors. Interestingly, however, these cytochrome $bcc$ hypomorphs were highly sensitized to disruptors of membrane potential or cAMP levels (Supplementary Fig. 11b).

PCLs derived from phenazine-like molecules including clofazimine (Fig. 3c) also showed sensitization of the Dxr and IspD hypomorphs. The precise mechanism of action of clofazimine and other phenazines has been debated; although interference with NADH dehydrogenase activity has been reported[72,73], phenazines have been shown to retain activity in $Mtb$ strains lacking Ndh-2[74]. The sensitivity of the Dxr and IspD hypomorphs in PROSPECT suggests that interference with the reduction of menaquinone by NDH is likely a critical feature of phenazine activity. Despite sharing some important common features, the behavior of the rest of the strains in the pool, in particular the MenC and EntC hypomorphs, enable the successful separation of phenazines from QcrB inhibitors using PCLs.

## Experimental validation of PCL respiration predictions for unannotated GSK set compounds
To obtain experimental validation of the PCL respiration predictions, we measured MIC (minimum inhibitory concentration that inhibits 90% of bacterial growth) against two non-PROSPECT mutants commonly used to confirm QcrB inhibition (Fig. 3d, Supplementary Fig. 11c)[18]. One mutant was a $cydA$ loss of function mutant, which is expected to show enhanced sensitivity to QcrB inhibitors due to the ability of cytochrome oxidase ($cydA$) to compensate for QcrB inhibition[18]. The second mutant carries a QcrB A317T allele that has been shown to confer resistance to a broad range of QcrB inhibitors[18]. Thus, hypersensitivity of the $cydA$ mutant and resistance of the $qcrB_{A317T}$ mutant to a compound would indicate QcrB as the target. Activity in these two strains also served to differentiate QcrB inhibitors from other respiration inhibitors, like phenazines such as clofazimine, which show no such corresponding shift in activity against the two mutants.

We tested 29 of the unannotated GSK compounds that PCL analysis newly predicted to target respiration against these two QcrB MOA identifying mutants: 23 newly predicted QcrB-targeting molecules and 6 newly predicted phenazine-like molecules. 83% (19/23) of the putative QcrB inhibitors indeed exhibited shifts in sensitivity in the $cydA$::Tn and $qcrB_{A317T}$ mutants as would be expected for QcrB inhibitors. Newly identified QcrB inhibitors included two new scaffolds, napthalene carboxamides (NC; 2 compounds) and triazolothiadiazoles (TTDZ; 3 compounds) (Fig. 3e, f), along with 14 new structural singleton compounds (Supplementary Fig. 11a). The validated singletons include TCMDC-142962 (GSK1107112A), a compound previously shown to bind and inhibit both EthR and InhA[40], although neither of these activities was sufficient to account for its whole-cell activity. PROSPECT and PCL analysis thus correctly assigned the MOA of these 19 unannotated compounds despite their lack of structural similarity to any of the known QcrB inhibitors in the reference set (Supplementary Data 6, Supplementary Note 13). We also tested Q203 and 32 of the previously annotated GSK QcrB inhibitors that all showed the expected shifts in MIC to the mutants (Supplementary Fig. 11c).

Conversely, for the available compounds predicted to a phenazine-like PCL, neither mutant showed >2X MIC shift relative to H37Rv (Fig. 3d), as is observed with clofazimine (Supplementary Fig. 11c), consistent with the PCL-analysis predictions. Overall, the phenazine-like predicted compounds included 2 structural groups, diphenylamines (DPA) and 2-anilopyridines (2-AP) (Fig. 3g, h) and 3 singletons (Supplementary Fig. 11a, Supplementary Data 5).

### Prioritization and validation of a novel QcrB inhibitor from primary PROSPECT screening data

We had previously performed primary, single-dose (50 μM) PROSPECT screens of ~104,000 total compounds from unbiased chemical libraries, i.e., collections that had not been preselected for anti-tubercular activity[8]. We selected 5146 compounds based on potency and/or evidence of strain specificity in their CGI profiles. Many of the selected compounds were without initial wild-type H37Rv activity. We then performed PROSPECT in 10-point dose response on this set, as described above for the reference set, and applied PCL analysis to predict MOA. To demonstrate the utility of our method for prioritizing molecules of interest with no wild-type activity, we picked a wild-type-inactive compound from this set that had a high-confidence MOA prediction to QcrB, BRD4310, a pyrazolo[3,4-*d*]pyrimidine (Fig. 4a). The CGI profiles of BRD4310 matched with high confidence to 3 QcrB PCLs at 3 different doses (Supplementary Fig. 12a). We thus chose this compound for a detailed validation of our PCL-derived QcrB prediction from screening of unbiased chemical libraries.

The dose-response PROSPECT data for BRD4310 showed minimal wild-type activity at concentrations below 50 μM (Fig. 4b). However, medicinal chemistry efforts generated an analogue, BRD4310a, with a low MIC value (5.7 μM) against wild-type *Mtb* strain H37Rv (Fig. 4a, Supplementary Note 14). BRD4310a demonstrated activity against *M. bovis* bacille Calmette–Guerin (BCG), and *M. marinum*, but not *M. smegmatis*, *M. fortuitum*, or *M.abscessus* or gram-positive *Staphylococcus aureus* and *Enterococcus faecalis* or gram-negative *Klebsiella pneumoniae*, *Escherichia coli*, and *Pseudomonas aeruginosa* (Supplementary Fig. 12b). BRD4310a was bacteriostatic in H37Rv (Supplementary Fig. 12c) with an $MBC_{90}$ (minimum bactericidal concentration) 16-fold higher than the MIC, consistent with QcrB inhibition. Similarly, BRD4310a had an 8-fold lower MIC against the *cydA* transposon insertion mutant relative to wild-type H37Rv, again consistent with cytochrome *bcc-aa3* inhibition (Fig. 4c).

We confirmed that QcrB is indeed the target of BRD4310a using genetic and functional studies. We generated spontaneous resistant mutants to BRD4310a (frequency of ~$10^{-8}$). Whole-genome sequencing (WGS) of four independent clones revealed missense mutations in the *qcrB* gene (Fig. 4d). Specifically, resistant mutant (RM) 1 and 2 harbored Y161C and M310T mutations, respectively, yielding a 4-fold and 8-fold increase in MIC, compared to the parental strain. RM3 and RM4 had mutations resulting in an A317T amino acid change often described to confer QcrB inhibitor resistance. RM4 had an additional L176P change which has also been found to confer resistance to the QcrB inhibitor Q203, resulting in a significant MIC shift ( >12-fold) compared to the wild-type parent[75]. Indeed, all BRD4310a-resistant strains showed various levels of cross-resistance to Q203 (Supplementary Fig. 12d). Expression profiling of *Mtb* exposed to BRD4310a also confirmed a gene expression pattern similar to Q203 exposure (Fig. 4e, Supplementary Fig. 12e) with upregulation of the *cyd* operon (log2-fold change [log2FC] ≥2, adjusted P-value [$P_{adj}$] of ≤0.05), indicating compensation by cytochrome *bd* of the electron transport chain. Finally, we confirmed that BRD4310a indeed triggered rapid intracellular ATP reduction, similar to the effect of exposure to Q203, as expected for a QcrB inhibitor (Fig. 4f). Taken together, the microbiological, genetic and functional data confirmed the identification using PCL analysis of a novel pyrazolo[3,4-*d*]pyrimidine scaffold targeting QcrB directly from PROSPECT screening of an unbiased library, initially without potent wild-type activity but subsequently achieved by chemical optimization.

## Discussion

PROSPECT is a systems chemical biology strategy that expands the target space and chemical diversity of antimicrobial discovery by substituting a single wild-type strain with a pool of genetically engineered hypomorphs in high-throughput whole-cell screening. Here we present PCL analysis, a method for MOA inference directly from primary PROSPECT screening data. Being reference-based, this method enables the identification of MOA for compounds that act at established targets or pathways while highlighting compounds with novel MOAs because their CGI profiles deviate from all patterns within the reference set. We assembled a reference set of compounds with known, annotated MOA. Within each MOA in the reference set, we identified PCLs, namely groups of similar CGI profiles, to which we compared CGI profiles of unknown compounds to predict their MOA. When we applied PCL analysis to this reference set in a leave-one-out cross-validation setting, we estimated the sensitivity and precision of PCL analysis predictions to be 70%, and 75%, respectively. When applied to an external test set of published, annotated GSK compounds, the sensitivity and precision were 69% and 87%, respectively. We also made high-confidence MOA predictions for 61% of the remaining high value, potent and non-toxic GSK compounds that were previously unannotated compounds and experimentally confirmed with 83% accuracy a large subset of predicted QcrB inhibitors. Finally, we also described one compound, BRD4310, that was identified from PROSPECT screening of an unbiased library (i.e., not biased for anti-tubercular activity) which was inactive against wild-type *Mtb* but was identified because of the greater sensitivity of PROSPECT over wild-type screening. As PCL analysis predicted the scaffold to target QcrB, we demonstrated the ability to chemically optimize the scaffold for wild-type activity while biologically confirming the PCL prediction. Taken together, this study is proof of concept for a high-throughput strategy to yield new, potent antitubercular compounds with annotated MOA that might otherwise elude discovery, with early MOA prediction to guide hit prioritization.

As with any reference-based analysis[35,76], PCL is limited by the attributes of the reference set, including the breadth of mechanistic diversity, the density of compounds representing any given MOA, and the diversity of scaffolds sharing an MOA. In assembling the reference set, we favored published compounds with supporting biological data. However, we also included some compounds with less confident MOA annotation, including some with computationally predicted MOAs, to expand the breadth of MOAs and scaffolds represented, cognizant of the varying degrees of confidence one can have for any given annotation. Accordingly, the accuracy of PCL prediction was much greater for the subset of our reference set (170 compounds) with strong published evidence i.e., whole-cell mechanistic support and known antitubercular activity (90% precision, 88% sensitivity in LOOCV) than for the subset (73 compounds) where evidence for their MOA annotation was limited to target-based enzymatic assay or computational docking or mechanistic support only in other organisms with no reported whole-cell antitubercular activity (33% precision, 27% sensitivity) (Supplementary Table 4, Supplementary Data 4). Populating the reference set with enough compounds within individual MOAs and representatives of different MOAs is important for both positive association – identifying a PCL similar enough to a test CGI profile to define a confident prediction – but also for defining the boundaries of confidence, since the denser the mechanism space, the better the resolution and the more accurate the predictions. The ability of PROSPECT and PCL analysis to predict MOA will only improve as the reference set expands to include compounds with new MOAs, new

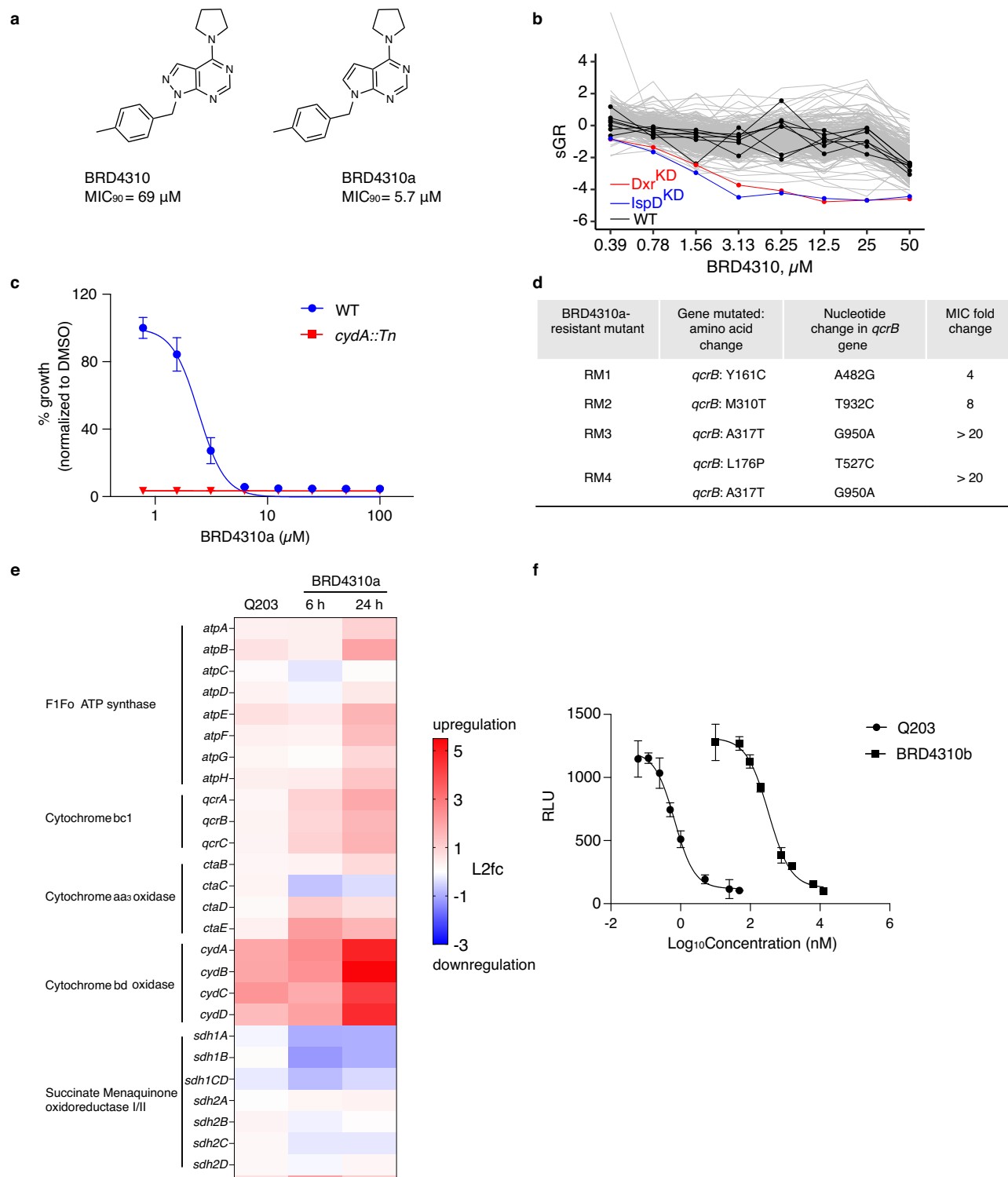

**Fig. 4 | Identification of a QcrB inhibitor from an unbiased library. a** Structures of screening hit BRD4310 and a synthesized analog, BRD4310a, with improved activity. **b** The chemical-genetic interaction (CGI) profiles for BRD4310 tested in dose-response, with Dxr (red) and IspD (blue) hypomorphs highlighted. Wild-type (WT) H37Rv strains are highlighted in black. **c** Sensitization of *cydA*::Tn upon BRD4310a treatment. **d** Mutations in BRD4310a-resistant H37Rv mutants identified by whole-genome sequencing. **e** The transcriptional responses of respiration-related genes in H37Rv to BRD4310a exposure at 6 and 24 h are similar to those observed with Q203 exposure at 6 h. The color scale indicates differential regulation as log2-fold change (L2fc) of H37Rv with compound condition relative to time-matched vehicle controls. Upregulation is indicated in red, downregulation is in blue. Data are from two independent RNAseq experiments each with 3 replicates (*n* = 3). **f** BRD4310a triggers the depletion of ATP levels of *Mtb* grown in aerobic conditions; Q203 was used as a control. Data from three independent experiments are presented as the mean ± s.d. Source data are provided as a Source Data file.

scaffolds that work by existing MOAs, and improved annotation of existing molecules.

Although the current screening pool covers most of the essential gene space (i.e., potential targets) in *Mtb*, it is not yet comprehensive. Thus, some known targets (e.g., InhA) or potential novel targets may not be represented by a corresponding hypomorph in the screening pool. Also, even if present, the direct molecular target of a compound is not always the most sensitized hypomorph due to factors including variable target vulnerability[6], variable levels of knockdown between strains, or the existence of highly vulnerable nodes in pathways upstream of the targeted process. Nevertheless, even in such cases, PCL analysis can assign MOA to molecules based on high CGI profile correlation with the CGI profile patterns of known compounds with the same target, rather than on individual strain behaviors. In fact, in some cases, non-target sentinel strains can drive CGI profile correlations and thus play central roles in identifying an MOA, rather than the actual corresponding target hypomorph. For example, the sensitivity of the Dxr and IspD hypomorphs to respiration inhibitors including QcrB inhibitors likely reflects the importance of isoprenoids at multiple steps in the electron transport chain, including a potentially competitive interaction at QcrB.

We applied PROSPECT with PCL analysis both to evaluate compounds with known potent wild-type antitubercular activity and to screen an unbiased library to discover new scaffolds, either with or without wild-type activity. In the first case, we evaluated compounds that had emerged from two screening campaigns at GSK, which evaluated collectively 2.25 million compounds and resulted in the prioritization of 227 compounds based on potency and lack of cytotoxicity[16,17]. Here, in a single screen and analysis of the 173 available compounds, we made high-confidence MOA predictions for 120 of them. Over a third of the compounds have had their MOAs published with independent elucidation since their release a decade ago, and these MOA assignments are largely concordant with our PCL predictions (87%).

Meanwhile, PCL analysis made novel MOA predictions for 60 compounds that previously lacked annotation, covering 10 different MOAs. These novel assignments revealed 19 new structurally diverse QcrB inhibitors that we subsequently experimentally validated, 9 molecules that behaved similarly to respiration-linked phenazines, 2 new sulfonamide adducts predicted to target DHPS, 9 compounds predicted to target DNA replication/nucleotide metabolism, and 16 predicted cell-wall active compounds. Of note, these PCL-based MOA predictions were made blinded to chemical structure; the broad structural diversity of the novel, validated QcrB inhibitors underscores the value of chemical-genetic interaction-based MOA prediction, as these predictions would not have been possible based on chemical similarity alone[77]. Indeed, as a naive baseline comparison to PROSPECT and PCL analysis, if MOA was simply inferred post hoc based on highest Tanimoto similarity to the reference set it would have only correctly assigned 35 of the 75 annotated GSK compounds, 1 of the 19 newly predicted and validated QcrB inhibitors, and failed to predict BRD4310 to inhibit QcrB (Supplementary Table 2, Supplementary Data 6, Supplementary Note 13).

These new predictions for previously unannotated GSK compounds now offer the opportunity to direct more focused study to more definitively elucidate the MOAs of specific compounds and perhaps pave the way for the development of these potent antitubercular compounds. Despite their working by established MOAs, these molecules are structurally distinct from the reference compounds upon which their predictions are based (Fig. 3e-h, Supplementary Fig. 11a, Supplementary Data 6), suggesting that in some cases, they could potentially overcome existing antibiotic resistance to reference inhibitors. Alternatively, PCL analysis can rapidly associate a hit compound with an MOA that may be of less interest, enabling its rapid de-prioritization and investment of effort elsewhere. Finally, an

important corollary is that PCL analysis also enables the recognition of compounds that do not act like any of the compounds in the reference set, and thus potentially work by novel mechanisms that may be of significant interest.

Among the 173 GSK compounds that were available for testing, PCL analysis predicted a remarkably high number of them to be QcrB inhibitors (65; 38%). While an enrichment for QcrB inhibitors is not altogether surprising given the number of different QcrB inhibitors that have now been reported from both within and outside this set (reviewed in ref. 78), this systematic analysis provides dramatic support for the anecdotal observation of the high rate of discovery of QcrB inhibitors emerging from whole-cell screening on a case-by-case basis in the literature. The well-established approaches to MOA identification for whole-cell active molecules targeting QcrB, including the multiple QcrB mutant alleles conferring broad resistance to structurally diverse inhibitors, could possibly contribute to a biased, unusually high number of literature reports of new QcrB inhibitors, given the confounding prerequisite of target elucidation typically required for publication. This work however, unencumbered by this prerequisite, suggests a true enrichment of QcrB inhibitors among moderate to highly potent whole-cell active compounds ($<10 \mu M$) with low toxicity, the criteria for inclusion in the GSK published sets. While the moderate chemical diversity of the analyzed 173 GSK compounds may also dampen the dramatic statistics − 112 compounds share significant structural similarity to at least one other compound in the set, resulting in 29 structural groups[79] ranging in size from 2 to 16 (Supplementary Data 5) with 61 structural singletons − the overrepresentation of QcrB inhibitors (22 of these 91 chemotypes) nevertheless holds. It should be noted that media conditions in the original GSK screen and PROSPECT were somewhat favorable for the discovery of compounds with this MOA, as neither assay included glycerol as a carbon source, which has been shown to limit the activity of QcrB inhibitors[80].

It then remains an interesting question as to why there is such a high prevalence and diversity of QcrB-inhibiting scaffolds discovered with whole-cell activity, which may point to some fundamental chemical biological principles of the vulnerability and druggability of protein targets. The case study of whole-cell screening for antibacterial activity in a bacterium with potentially 600 possible essential targets and the apparent bias for a small number of specific targets (e.g., QcrB, MmpL3, DprE1) calls for understanding the vulnerability of these targets as the basis for this bias. In the case of QcrB, the underlying basis is made even more intriguing by the cross-resistance to numerous, structurally diverse compounds conferred by the same, small handful of resistance mutations. At an even higher level, that this systematic study predicts with high confidence that 69% of the 173 analyzed molecules work by known MOAs (i.e., there are pre-existing, known molecules that work by the same MOA) reinforces the need to understand this repeated, preferential discovery of inhibitors of the same targets and pathways over others, particularly in the context of the desire to find inhibitors with novel MOAs.

Finally, by applying PCL analysis directly to PROSPECT screening data of an unbiased library, we identified a new pyrazolo[3,4-*d*]pyrimidine scaffold working by an established MOA (QcrB inhibition). The initial molecule had limited wild-type activity at the screening concentration of 50 μM but clear, discernible activity against the Dxr and IspD hypomorphs, even at sub-micromolar concentrations, thereby enabling not only the identification of an active molecule that would have eluded discovery by conventional wild-type screening, but also early, direct assignment of a putative MOA that can be integrated into hit prioritization. In contrast, conventional whole-cell screening strategies frequently assign MOA late, often only after extensive chemistry efforts to synthesize even more potent molecules have been invested to enable target identification. In many cases, this effort has resulted in disappointment when the target is eventually revealed to be of low

interest, or the target is never identified. Here, we advanced the potency of this scaffold to achieve low micromolar wild-type activity while confirming the predicted PCL target, thus showing again that the greater sensitivity of PROSPECT for compound activity can significantly expand the active chemical space to yield new lead anti-tubercular compounds that would not have been discovered by any conventional strategy, with accurate early MOA prediction.

By providing insight into MOAs directly from primary screening data, PROSPECT with PCL analysis is poised to fundamentally change antibiotic discovery by enabling the early integration of biological insight with varying degrees of resolution, ranging from actual target prediction to simply recognizing the novelty of its MOA, with the traditional metrics used to prioritize candidates emerging from screening campaigns, such as potency and chemical features. Even while better computational methods are needed to address the challenge of MOA predictions in the case of molecules with completely novel mechanisms in a reference-free manner, PROSPECT with PCL analysis can nevertheless rapidly identify new scaffolds with known MOAs and address the problem of rising antibiotic resistance in the face of an inadequate antimicrobial development pipeline.

## Methods

### Bacterial strains and culture conditions

*Mtb* H37Rv and derivative strains were cultured in Middlebrook 7H9 liquid medium supplemented with 10% oleic albumin dextrose catalase (OADC), 10 mM sodium acetate and 0.05% Tween 80. Anhydrotetracycline (500 ng/ml), hygromycin (50 μg/mL), and/or streptomycin (20 μg/mL) were used when required. For solid medium, Middlebrook 7H10 agar medium supplemented with OADC and 0.5% glycerol was used. The hypomorph strains were generated using a protein degradation system that has been reported[9]. The *cydA*::Tn was isolated from an arrayed PhiMycoMar-generated transposon library; the MycoMar transposon is inserted at the TA following codon G12 in *cydA*. For Supplementary Fig. 12, the following strains were used: *M. bovis* bacillus Calmette–Guérin Pasteur, *M. marinum* strain M, *M. smegmatis* mc²155, *M. abscessus* ATCC 19977, *M. fortuitum* ATCC 6841, *Escherichia coli* MG1655, *Pseudomonas aeruginosa* (PAO1), *Staphylococcus aureus* Newman, *Klebsiella pneumoniae* ST258 and *Enterococcus faecalis* RB027[81].

### PROSPECT screening of compound libraries

PROSPECT screening[8] is a 14-day outgrowth assay in 384-well plates, using Middlebrook 7H9 supplemented as described above, with dextrose and acetate serving as primary carbon sources. Prior to the screening, each of the hypomorphic strains is grown independently to mid-log phase in anhydrotetracycline (ATc) so that SspB is repressed[9]. On the day of screening, the cultures are washed and resuspended in an ATc-free medium to induce the hypomorphic phenotype and combined at an overall A600 of 0.0075 prior to transfer into assay plates. On every assay plate, columns 2 and 23 contained alternating DMSO (vehicle control) and 256 nM or 498 nM rifampin (positive control), whereas rows A, B, O, and P and columns 1 and 24 were left empty. Assay plates were cultured for 14 days at 37 °C. Assay plates were heat-killed at 80 °C for 2 h, the bacteria lysed, and the barcodes were amplified for Illumina sequencing. PCR libraries were combined, SPRI purified, and sequenced at the Broad Institute Genomics Platform using Illumina HiSeq 2500 at an average sequencing depth of at least 500 reads per strain per well.

The primary, single-dose screen that initially identified BRD4310 used a pool of 155 strains, including a single barcoded H37Rv control. Screening of the reference set, the blinded GSK compounds, BRD4310 in dose-response, and the rest of our compound libraries occurred over six separate waves of screening using larger pools of 396 up to 466 strains including 7 barcoded variants of wild-type H37Rv. Two spiked-in barcodes were included that serve as controls. One spike-in

(tag-8090, i.e., Lysis-control) was included during the cell lysis step at a concentration of 0.172 pg/mL in the 20% (v/v) aqueous DMSO that was added to each well prior to heating at 98 °C for 10 min. The other spike-in (tag-1180, i.e., PCR-control) was added at 0.354 pg/mL to the prepared PCL Q5 Master Mix for inclusion at the PCR step.

### Data processing and analysis of PROSPECT data

Following next-generation sequencing, strain barcode counts were deconvoluted from raw FASTQ files using the ConCensusMap script[8]. To correct for technical variability arising from PCR primers and batches, counts were log2-normalized based on the median abundance of spike-in control barcodes. Log2-fold change (L2FC) for each strain relative to vehicle controls was then calculated using the ConcensusGLM package in R. Growth rate (GR) scores were calculated from L2FC values by normalizing to onboard positive (rifampin) and negative (vehicle) controls, as adapted from Hafner et al.[26]. To mitigate PCR jackpotting effects, GR values > 5 were scaled down using log10-scaling. Strains that exhibited slow growth (L2FC in rifampin > −1), unreliable growth (GR > 20), or that were not present in all screening waves were filtered, yielding 340 strains for downstream analysis. To enable comparison across all strains and conditions, GR values were standardized by first performing quantile normalization and then calculating a robust z-score to generate the standardized growth rate (sGR) using the CmapM MATLAB package (https://github.com/cmap/cmapM[26,27,33,82]). The resulting sGR profiles were used for all further analyses. To visualize the relationships between conditions, the reference set sGR profiles were embedded in two dimensions using UMAP implementation in the R package UMAP, based on the Pearson distance between conditions. Full methodological and theoretical details are provided in Supplementary Notes 1–5.

### Reference set assembly

The reference compound set was assembled based on published reports (Supplementary Data 1). Literature evidence for compounds' mechanisms of action was reviewed in PubChem[83], CAS SciFinder[84], and DrugBank[85] and updated up to the point of PCL analysis to maintain the most accurate and up-to-date reference set. Due to limited commercial availability, analogs of the exact molecules shown and discussed in some publications were instead acquired and used as reference. MOA was annotated as specifically as possible to target protein where published evidence existed for specific inhibition and in other cases to multi-subunit protein complex (e.g., 50S ribosomal subunit, 30S ribosomal subunit) and biochemical pathway in TB (e.g., purine metabolism and pyrimidine metabolism for nucleoside analogs with multiple proposed enzymes that they can competitively inhibit). The reference set included some compounds with published activity against more than one protein which were included in more than one annotated MOA reference group in the PCL analysis. Strength of evidence for a particular MOA differed between compounds from protein targets validated via co-crystal structures, strains with mutations in the target protein conferring resistance, enzymatic inhibition in biochemical assays, computational docking, and target evidence so far only in other bacteria or eukaryotic cells which could reasonably limit our ability to predict some annotated MOAs in this assay.

### Screening compounds

Compounds from the reference set were tested in PROSPECT in duplicate in a 10-point series of concentrations ranging from 0.1 μM to 50 μM with 2-fold dilution. When required, highly potent compounds had their dose series adjusted based on their MIC. Compounds kindly provided by GSK were first tested in dose response in a 7-day outgrowth assay using an H37Rv strain constitutively expressing GFP[86]. Based on these data, compounds were screened in PROSPECT across an 8-point two-fold dilution series (in duplicate) tailored to start at 2X

the observed MIC for each compound. For compounds that did not show clear wild-type activity under our conditions, the dilution series started at a maximum concentration of 50 μM. BRD4310 was screened in duplicate in an 8-point two-fold dilution series with the maximum concentration of 50 μM.

## Clustering of sGR profiles and PCL similarity scoring

Briefly, to cluster sGR profiles and define predictive similarity scores (see Supplementary Notes 7–8 for full details), we first applied spectral clustering in MATLAB (https://www.mathworks.com/help/stats/spectralcluster.html[31,32,87,88]) to the standardized growth rate (sGR) profiles within each MOA category. A nearest-neighbor graph was constructed for each MOA based on Pearson correlation, and the number of clusters, $k$, was automatically estimated using an eigengap heuristic on the graph Laplacian. The spectralcluster function was then used to partition the graph into $k$ clusters. We defined and calculated a PCL similarity score between each test CGI profile and each cluster from unsupervised, spectral clustering as the median of the Pearson correlation coefficients between the test CGI profile and all reference set CGI profiles within the cluster. Clusters whose highest similarity scoring, reference set CGI profile shared the same annotated MOA as the cluster members (i.e., one of the cluster CGI profiles themselves or a CGI profile from a compound in the same MOA but from a different cluster) were kept for use in making MOA predictions and defined as PCLs.

## MOA prediction using PCLs and validation

We approached MOA prediction as a multi-class classification problem using a one-vs-rest (OvR) strategy. To predict the MOA of unknown compounds, we trained a one-vs-rest (OvR) classifier for each PCL cluster, using reference compounds annotated with the same MOA as positive examples and all others as negatives. Each classifier mapped a reference compound dose profile's similarity score to the PCL to a confidence score representing the positive predictive value (PPV) of MOA assignment at that threshold using the perfcurve function in Matlab (https://www.mathworks.com/help/stats/perfcurve.html[89–91]). For unknown compounds, CGI profile similarity scores were used to linearly interpolate and estimate PCL confidence scores from the reference set mappings. MOAs were predicted based on the highest-confidence PCL across all profiles, using a majority-rules vote across doses to break ties. High-confidence predictions (PCL confidence score = 1) were distinguished from lower-confidence or uncertain calls. The performance and robustness of this entire framework were rigorously evaluated using leave-one-out cross-validation (LOOCV), where each reference compound was excluded in turn and MOA predicted using PCL clusters and their classifiers trained only on the remainder. MOA predictions were compared against each compound's annotated MOA to calculate micro-averaged precision, sensitivity, and F1 score. We further benchmarked our approach by comparing its performance to simpler methods, such as using whole MOAs instead of clusters from spectral clustering, applying a single ROC-derived similarity threshold for all PCLs, or using a 1-nearest-neighbor classifier. Full methodological and theoretical details are provided in Supplementary Notes 9–12.

## Selection and sequencing of resistant mutants

The generation of BRD4310a-resistant mutants was adapted from a previously established method[19]. Briefly, the log-phase H37Rv cultures were plated onto 7H10 agar plates containing 2×, 4×, and 8× MIC. Approximately $10^8$ CFU of *Mtb* was spread per plate. After 6 weeks of incubation at 37 °C, the colonies were picked and grown in 7H9 medium containing 2xMIC of BRD4310a. The cultures were retested in MIC assay to confirm the resistance. Genomic DNA of the BRD4310a-resistant mutants was isolated. Paired-end libraries were prepared using Illumina Nextera XT DNA library preparation kit and sequenced

on Illumina HiSeq platform. Sequencing reads were mapped to the AL123456 reference genome and single nucleotide polymorphisms (SNPs) were called using Pilon[92]. Only SNPs unique to the resistant strains and absent from the parent strains are reported. The mutations identified in *qcrB* by whole-genome sequencing were subsequently confirmed by Sanger sequencing.

## Determination of minimum inhibitory concentration (MIC) and minimum bactericidal concentration (MBC90)

Bacteria were grown to mid-log phase, then 100 μL of mycobacteria culture diluted to $OD_{600}$ 0.0025 was dispensed into each well on 96-well flat-bottom plates containing an 8-point, two-fold dilution series of compound, with concentrations tailored to the known activity of the compound, delivered to assay plates in a constant 1 μL volume of DMSO. The assay plates were incubated for 7-day at 37 °C. The effect of the compound-dose condition on the bacteria was assessed by OD600 using a Spectramax M5 plate reader. MIC values were defined as the drug concentration that inhibited 90% of bacterial growth relative to control wells. MIC was assessed by fitting a 4-parameter curve to dose-response data after normalization to on-plate controls (DMSO = no inhibition, 100 nM rifampin = 100% inhibition, $n = 6$ per plate for each control). All compound dilution series were assayed on two replicate plates, and dose-response curves were fitted to the normalized replicate data points. In testing against $qcrB_{A317T}$ and *cydA*::Tn mutants, apparent MICs were occasionally outside of the range of concentrations tested; because available compound volumes were limited, in these cases, the log2-fold change shown in Fig. 4 represents a minimum estimate, using the maximal concentration tested as the MIC for the calculation of the MIC shift for the $qcrB_{A317T}$ mutant or the minimum concentration tested for the *cydA*::Tn mutant. For determination of bactericidal activity, aliquots of similarly cultured bacteria were plated onto 7H10 agar plates to determine colony-forming units (CFU) at the time points indicated, and colonies were counted after 3 weeks of growth.

## RNA isolation

H37Rv cultures were grown to an $OD_{600}$ of 0.4. Compounds were added as indicated: 20 nM Q203 and 25 μM BRD4310a. DMSO was used as a vehicle control. Culture samples were harvested in triplicate at the appropriate time points. 2 ml of culture were pelleted, and the supernatant was decanted. The cell pellet was resuspended in 0.5 mL of TRIzol (Invitrogen) and stored at −80 °C overnight. RNA was purified using Direct-zol RNA Kits (Zymo research) and quantified with Nanodrop spectrophotometer (ThermoFisher).

## RNAseq experiment

Illumina cDNA libraries were generated using a modified version of the RNAtag-seq protocol and sequenced on an Illumina NovaSeq platform[93]. Sequencing reads from each sample in a pool were demultiplexed based on their associated barcode sequence using custom scripts (https://github.com/broadinstitute/split_merge_pl). Up to 1 mismatch in the barcode was allowed provided it did not make assignment of the read to a different barcode possible. Barcode sequences were removed from the first read as were terminal G's from the second read that may have been added by SMARTScribe during template switching. Reads were aligned to NC_000962 using Burrows-Wheeler Alignment and read counts were assigned to genes and other genomic features using custom scripts (https://github.com/broadinstitute/BactRNASeqCount). Differential gene expression analysis of H37Rv was conducted with DESeq2 with compound condition relative to time-matched vehicle controls.

## ATP assay

Bacterial ATP was measured using the BacTiter-Glo Microbial cell viability assay (Promega). Briefly, in 96-well plate varying concentrations

of compounds were dispensed, then 100 µL of H37Rv culture was added to each well. The assay plates were incubated at 37 °C for 24 h. BacTiter-Glo reagent was added to each well and incubated further in the dark for 10 min. The luminescence was recorded using the Spectramax M5 plate reader.

## Statistics & reproducibility

No statistical methods were used to predetermine sample size. The experiments were not randomized, and investigators were not blinded to allocation during experiments and outcome assessment.

For each screening wave independently, strains that grew slowly (did not achieve at least one doubling over the 14-day assay) or unreliably (any conditions with GR > 20) were identified and filtered out from further analysis. Of the 387 strains that were included in the strain pool across all six screening waves, 42 strains were filtered out for slow growth in at least one of the screening waves and 5 strains were filtered out for high GR; the 340 strains that passed quality control in all six screening waves were used in downstream analysis. Otherwise, no data were excluded from the analyses presented.

Sample sizes (n = 2) for screening were chosen as standard for high-throughput compound screening as a balance of cost and accuracy. The PROSPECT screening assay was previously optimized to allow 2 replicates to provide statistical power as described in Johnson et al. 2019. Primary data were generated in at least duplicate and were shown to give similar results (Supplementary Fig. 5b, Supplementary Note 4). Results were confirmed using orthogonal methods, which demonstrated the reliability of the primary data as described. Follow-up mechanism of action studies were replicated at least 3 times. Due to limited compound availability, testing of GSK compounds against *cydA::Tn* and *qcrB* mutants was performed in duplicate. All attempts at replication were successful.

Investigators were not blinded during data collection or analysis for reasons of feasibility. Compounds in our study were assigned ID numbers, essentially blinding their identity until after collection and analysis was complete. All PCL-based MOA assignments were based exclusively on CGI profiling and were blinded to compound structure or other associated information. Follow-up communication with GSK unblinded the chemical identities of the compounds and allowed for literature review of reported mechanistic activities. For MIC assays, to eliminate errors introduced during compound dilution, assays using different *Mtb* strains were performed on the same day, using the same DMSO-based compound dilution series. Due to limited compound availability, MIC assays to investigate potential GSK QcrB-targeting compounds were performed once, in replicate, as described. In cases where the curve fit MIC fell outside of the range of concentrations assayed, the minimal or maximal assay concentration was used to calculate the magnitude of the MIC shift; in these cases, the actual shift is likely greater than the shift indicated in Fig. 3d.

## Reporting summary

Further information on research design is available in the Nature Portfolio Reporting Summary linked to this article.

## Data availability

The standardized growth rate (sGR), Pearson correlation to reference CGI profiles, average rank of Pearson correlation across reference CGI profiles, reference CGI profile PCL cluster membership, PCL similarity score, and PCL confidence score data for reference CGI profiles, the GSK set, and BRD4310 are available online on Code Ocean within the published code capsule at https://doi.org/10.24433/CO.3013890.v1 and have been deposited in Figshare: https://doi.org/10.6084/m9.figshare.28373561. The reference set and GSK compound MOA annotations are available within Supplementary Data 1 and Supplementary Data 5, respectively. RNA-seq and resistant mutant whole-genome sequencing datasets were deposited in the Sequence Read Archive

(SRA) operated by the National Center for Biotechnology Information (NCBI) under BioProject accession number: PRJNA1328039. Source data are provided with this paper.

## Code availability

ConcensusGLM is available on GitHub at https://github.com/broadinstitute/concensusGLM. CmapM is available on GitHub at https://github.com/cmap/cmapM. CmapR is available through Bioconductor and on GitHub at https://github.com/cmap/cmapR. Computer code for running each step of the reference-based PCL analysis is available on Code Ocean (https://doi.org/10.24433/CO.3013890.v1) and GitHub (https://github.com/broadinstitute/Mtb_PROSPECT_PCL_analysis). Other computer code is available from the corresponding author upon request.

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

## Acknowledgements

We thank Rob Bates and GlaxoSmithKline for kindly providing the TB set compounds for both primary screening and follow-up studies. RNA-Seq libraries were constructed and sequenced by the Infectious Disease and Microbiome Program's Microbial Omics Core at the Broad Institute of MIT and Harvard. Funding for this work was provided by Bill and Melinda Gates Foundation (OPP1084233 D.T.H., INV-040933 D.T.H., INV-064678 D.T.H.), the Broad Institute Tuberculosis donor group, and the Pershing Square Foundation.

## Author contributions

M.O., S.Z., J.E.G., and D.T.H. conceived the study. The manuscript was written by A.N.B., S.Z., I.B.Z., N.S., J.E.G., and D.T.H. The computational analysis was designed by A.N.B., M.O., I.B.Z., N.S., J.E.G., and D.T.H. The computational pipeline was written and executed by A.N.B. and M.O. The reference set and GSK antitubercular set MOA annotations were curated by A.N.B., M.O., S.Z., K.L., M.F., D.K.H., J.E.G., and D.T.H. Experiments were designed as follows. Strain construction: M.F., S.E., E.J.R., C.M.S., D.S., J.E.G., D.T.H. Assay development and screening: M.F., J.E.G., D.T.H. Medicinal chemistry: K.L., D.K.H., D.T.H. Mechanism of action follow-up: S.Z., J.E.G., D.T.H. Experiments were carried out as follows. Strain construction: M.C., K.D., E.G., A.L.G., R.N. Compound screening: A.N.B., S.Z., M.C., K.D., E.G., A.L.G., R.N., N.G., J.E.G. Mechanism of action follow-up: S.Z., A.L., N.G., J.E.G.

## Competing interests

The authors declare no competing interests.

## Additional information

Austin N. Bond [1,8], Marek Orzechowski[1,8], Shuting Zhang[1,8], Ishay Ben-Zion [1], Allison Lemmer[1], Nathaniel Garry [1], Katie Lee[1], Michael Chen[1], Kayla Delano[1], Emily Gath[1], A. Lorelei Golas[1], Raymond Nietupski[1], Michael Fitzgerald[1], Sabine Ehrt [2], Eric J. Rubin [3], Christopher M. Sassetti [4], Dirk Schnappinger [2], Noam Shoresh[1], Diana K. Hunt [1], James E. Gomez [1] ✉ & Deborah T. Hung [1,5,6,7] ✉

[1]Broad Institute of MIT and Harvard; Cambridge, Massachusetts, USA. [2]Department of Microbiology and Immunology, Weill Cornell Medical College, New York, New York, USA. [3]Department of Immunology and Infectious Diseases, Harvard T. H. Chan School of Public Health, Boston, MA, USA. [4]Department of Microbiology, UMass Chan Medical School, Worcester, MA, USA. [5]Department of Molecular Biology and Center for Computational and Integrative Biology, Massachusetts General Hospital, Boston, MA, USA. [6]Department of Genetics, Harvard Medical School, Boston, MA, USA. [7]Center for Integrated Solutions in Infectious Diseases, Broad Institute of Harvard and MIT, Cambridge, MA, USA. [8]These authors contributed equally: Austin N. Bond, Marek Orzechowski, Shuting Zhang. ✉e-mail: jgomez@broadinstitute.org; hung@molbio.mgh.harvard.edu

