## [Transparent Peer Review file · Nature Communications]

Reference-based chemical-genetic interaction profiling to elucidate small molecule mechanism of action in *Mycobacterium tuberculosis*

Corresponding Author: Professor Deborah Hung

Version 0:

Reviewer comments:

Reviewer #1

(Remarks to the Author)

Bond and colleagues have submitted a manuscript for review entitled "Reference-based chemical-genetic interaction profiling to elucidate small molecule mechanism of action in *Mycobacterium tuberculosis*." The authors disclose the further refinement of their PROSPECT platform to the application of target identification for antitubercular small molecules. This method as demonstrated in the manuscript is significant to the TB field, and more generally to the antibacterial community, because of its demonstrated ability to afford hits and insights into their mechanism of action as provided by the CGI vector and the ensuing PCL analysis. The methodology shows value through the inference of mechanism of action with a set of GSK antituberculars.

The platform is demonstrated to have significant predictive ability as to the primary target of an antitubercular. With the analysis of the success of the predictions, are the authors able to correlate it with the "strength of evidence" for that target in the reference data set? This would be useful information for those in the field pursuing mechanistic studies related to new targets. For example, do docking results alone provide substantial target validation? In cases where a secondary target has been predicted, are the authors able to comment on the validity of the prediction?

A concern exists over the "Prioritization and validation of a novel QcrB inhibitor from primary PROSPECT screening data." This does not seem to be the best exemplification of this aspect of the platform given the publication of QcrB inhibitors. This reviewer would have liked to have seen the authors pick a molecule that is evidenced to inhibit a novel, or significantly understudied, target (perhaps one of significant interest due to a high vulnerability score but without published inhibitors). Otherwise, this reviewer is wondering why one would choose a compound without significant activity but mechanistic support for targeting QcrB. Are the authors able to provide a different example in this application of the methodology?

Overall, with the addressing of these concerns, this manuscript should be an excellent contribution to this journal.

Minor points:

- Extended Fig. 12 needs further information (e.g., strain number) to identify the strains
- All chemical reactions shown should, as per convention, depict all critical reagents above/under the reaction arrow.
- In chemical reaction procedures, please make corrections as to show the correct number of significant figures
- The chemical reaction procedures should list expected molecular formula and mass in addition to observed mass

(Remarks on code availability)

Reviewer #2

(Remarks to the Author)

The manuscript describes an informative and elaborate reference-based analysis and Mechanism-of-Action (MOA) prediction method applied to data generated using the PROSPECT method previously described by the authors.

PROSPECT analyzes the impact of drug candidates on a pool of hypomorphs, each of which is highly sensitive to small perturbations in the expression level of a certain gene. The sensitivities of individual hypomorphs to various drug candidates can then be used to derive Chemical-Genetic Interactions (CGI) profiles. Clustered CGI profiles enabled assignment of MOA with high precision and sensitivity to both established and novel compounds. It was of particular note that the authors were able to identify potential inhibitors by CGI profile comparisons even when the hypomorph of the direct target showed no sensitivity. Although, like most reference-based methods, PCL of the PROSPECT data is not able to derive novel MOAs with high confidence, it effectively reveals new chemical moieties that could evade known resistance mechanisms to previous treatments for Mtb. More importantly, the method enables the acquisition of exciting biological and mechanistic data for the compounds and the system organism, while traditional High-Throughput Screening (HTS) methods can only provide simple qualitative answers.

In general the data analysis is rigorous and informative and overall this is a really important contribution to the field of TB drug discovery. Below are a several points that the authors might consider addressing prior to publication.

1. A clear description of PROSPECT and PCL analysis in the main text would be helpful. For instance stating how growth rate is measured. In addition, PCL analysis is described but it is buried in supplementary notes 8 (and 7?).
2. In addition it would be helpful to describe how PROSPECT and CGI analysis deals with situations such as prodrugs, and compounds that directly target ClpXP. Could this be a source of some of the compounds whose MOA could not be predicted? For instance both thioacetazone (EthA) and isoniazid (KatG) are included in the data sets.
3. Along similar lines, can the authors speculate about the antitubercular leads with known MOA that they were not able to predict. Are these compounds clustered around specific targets and/or are there reasons why their MOA might be more complex or different from their 'known' MOA? The reference based approach depends on how well curated the existing 'known' compounds are and the PCL analysis is a great tool to fact check the reference compounds. In addition, it might also be possible to learn whether the success rate is higher for compounds with similar scaffolds to the reference compounds.
4. In the original manuscript the authors state that hypomorphs were generated either by conditional proteolysis (targeting proteins to ClpXP) or transcriptional control. What fraction of hypomorphs fall into each method of control? Is this a source of variability in the ease or difficult of predicting MOA?
5. The spectral clustering and PCL analysis is conducted using dose response curves. Is it possible to comment on how this increases the accuracy of the method? i.e. is the expectation that the MOA is concentration-dependent? Using only a single dose would reduce the dimensionality and complexity of the data and facilitate the analysis of other variables.
6. It looked like the Hill slope from the concentration response plots was allowed to float(?) If so does the Hill slope provide any additional insight into MOA and is it useful for clustering? Presumably compounds with the same MOA should have similar Hill slopes.
7. Only a single growth condition was used. This is clearly an important variable and it is obviously impossible to screen all compounds under all growth conditions known to affect target vulnerability. However, is it possible to speculate how the choice of growth condition might bias the results? For instance, could this partly account for the preponderance of QcrB actives?
8. Additionally, it would be great if the authors were able to comment on the singletons. It seems that there are singletons in the dataset that were derived from good quality measurements but weren't included due to the long distance from other compounds in the interaction map. Nonetheless, could these singletons represent completely novel MOAs? Maybe the authors could comment.
9. Three datasets of compounds were used - 437 from data mining, 173 from GSK, and >5000 compounds from the original PROSPECT paper. It would be nice to have a schematic cartoon clarifying all the connections between three datasets and outcomes of the analyses.
10. Historically HTS of compound series have also revealed multiple hits for MmpL3 and Drpe1 however QcrB is heavily represented in the output. Could the authors speculate/discuss why?
11. Please check references. For instance ref 22 in the following text should be 76.

To address this problem, we implemented a metric of (dose-dependent) compound-induced growth rate, GR, originally developed to measure the effect of a chemical perturbation on a collection of cancer cell lines exhibiting a range of proliferation rates²². This is not reference 22 but 76
12. Can the authors describe the 'optimization' of BRD4310 to BRD4310a? Improving cell-based activity is an important component of TB drug discovery. Are there any lessons to be learnt?
13. Please check the reported NMR chemical shifts for BRD4310a since a proton might be missing.
14. Please give chemical formulas for BRD4310 and BRD4310a.

15. Iron acquisition (EntC). Isochorismate is used for both mycobactin and menaquinone biosynthesis. Is the EntC gene described here in the gene cluster for mycobactin biosynthesis (as implied by 'iron acquisition') or for menaquinone biosynthesis? This section of the manuscript also discusses MenC which is in the menaquinone biosynthesis pathway. It would be useful if the authors could add a sentence of two about this.

16. MIC90 is normally taken to indicate the MIC of 90% strains tested.

(Remarks on code availability)

Reviewer #3

(Remarks to the Author)

(Remarks on code availability)

Reviewer #4

(Remarks to the Author)

(Remarks on code availability)

Reviewer #5

(Remarks to the Author)

The manuscript by et al presents an interesting and highly relevant concept to try and infer mechanisms of action of Mtb growth inhibition of libraries of putative inhibitors in a system-wide manner at a high level of confidence. After training their method on small sets, including those previously reported by GSK, they finally applied it to a large, unbiased library and discovered several new mechanisms attributed to compounds previously not known to be anti-tubercular, potentially extending the anti-TB drug pipeline. One of the most important results of the study is that the lack of confident predictions for a particular drug may be an indication of a novel mechanism of anti-TB activity. The paper is well-written and introduces a number of innovative concepts in high-throughput chemical biology data analysis. From a biologist's perspective however, the following concerns should be addressed:

Minor concerns

1. It is not clear what is the magnitude of variance in their proteolytic knock-downs, and what is biological impact of the same.
2. The potential impact of pleiotropic effects is not considered in data analyses. For example, INH may impact more than one pathway and therefore have multiple MOAs. Additionally, the inhibition of a particular pathway may require compensatory changes at the level of the DNA, RNA or protein for Mtb to survive, which could potentially be missed.

(Remarks on code availability)

Version 1:

Reviewer comments:

Reviewer #1

(Remarks to the Author)

I am satisfied with the authors' responses and modifications to the manuscript. I recommend this manuscript to be accepted for publication.

(Remarks on code availability)

Reviewer #2

(Remarks to the Author)

The authors have addressed my comments.

(Remarks on code availability)

Reviewer #3

(Remarks to the Author)

(Remarks on code availability)

Reviewer #4

(Remarks to the Author)

(Remarks on code availability)

Point by point responses (in blue) to reviewer comments

Reviewer #1:

We appreciate the reviewer's recognition that "the manuscript is significant to the TB field, and more generally to the antibacterial community" and "with the addressing of concerns, this manuscript should be an excellent contribution to this journal."

1. The platform is demonstrated to have significant predictive ability as to the primary target of an antitubercular. With the analysis of the success of the predictions, are the authors able to correlate it with the "strength of evidence" for that target in the reference data set? This would be useful information for those in the field pursuing mechanistic studies related to new targets. For example, do docking results alone provide substantial target validation? In cases where a secondary target has been predicted, are the authors able to comment on the validity of the prediction?

We do see that success was much greater for the subset of our reference set (170 compounds) with strong published evidence, i.e., whole-cell mechanistic support and known antitubercular activity (90% precision, 88% sensitivity in LOOCV) than for the subset (73 compounds) where evidence for their MOA annotation was limited to target-based enzymatic assay or computational docking or mechanistic support only in other organisms with no reported whole-cell antitubercular activity (33% precision, 27% sensitivity). We have added information to the reference set annotation table (**Supplementary Data 1**) categorizing their existing mechanistic strength of evidence and a mention in the text (lines 494-500) referring to a new table (**Extended Data Table 4**) that summarizes this dependence on the strength of evidence.

2. A concern exists over the "Prioritization and validation of a novel QcrB inhibitor from primary PROSPECT screening data." This does not seem to be the best exemplification of this aspect of the platform given the publication of QcrB inhibitors. This reviewer would have liked to have seen the authors pick a molecule that is evidenced to inhibit a novel, or significantly understudied, target (perhaps one of significant interest due to a high vulnerability score but without published inhibitors). Otherwise, this reviewer is wondering why one would choose a compound without significant activity but mechanistic support for targeting QcrB. Are the authors able to provide a different example in this application of the methodology?

While PROSPECT was conceived with the goal of diversifying antitubercular discovery by expanding target space, the focus of this particular manuscript is a specific application of PROSPECT – the rapid and accurate identification of novel scaffolds sharing a mechanism with known reference compounds. We chose to provide an in-depth description of a QcrB inhibitor not only because it was relatively easy to validate, but importantly, it allowed us to highlight a primary principle of our reference-based analysis: we can use a complex chemical-genetic interaction profile to predict MOA, even when the hypomorph corresponding to the direct target is not sensitized. Furthermore, we chose a compound without initial wildtype activity because it showcases the much-increased sensitivity of PROSPECT over traditional whole-cell screens, thus enabling the discovery of anti-tubercular compounds that would

have otherwise eluded discovery. The goal of identifying molecules hitting novel targets is the focus of a different, parallel effort and subsequent manuscript.

Minor points:

-Extended Fig. 12 needs further information (e.g., strain number) to identify the strains.

We have added the relevant strain numbers to the Methods section (lines 626-630).

-All chemical reactions shown should, as per convention, depict all critical reagents above/under the reaction arrow.

We have revised all chemical reaction schemes to depict key reagents as requested (**Supplementary Note 14**).

-In chemical reaction procedures, please make corrections as to show the correct number of significant figures

We have corrected all numerical values for significant figures (**Supplementary Note 14**).

-The chemical reaction procedures should list expected molecular formula and mass in addition to observed mass

We have added the expected molecular formula and calculated mass alongside the observed mass (**Supplementary Note 14**).

Reviewer #2:

We thank the reviewer for their recognition that “the data analysis is rigorous and informative and overall this is a really important contribution to the field of TB drug discovery.”

1. A clear description of PROSPECT and PCL analysis in the main text would be helpful. For instance stating how growth rate is measured. In addition, PCL analysis is described but it is buried in supplementary notes 8 (and 7?).

We have added descriptions of PROSPECT and the growth rate metric to the Results section (lines 169-172, 194) and to the Methods section (lines 654-672), and a summary of PCL analysis to the Methods section (lines 701-735).

2. In addition it would be helpful to describe how PROSPECT and CGI analysis deals with situations such as prodrugs, and compounds that directly target ClpXP. Could this be a source of some of the compounds whose MOA could not be predicted? For instance both thioacetazone (EthA) and isoniazid (KatG) are included in the data sets.

There are indeed specific unusual cases that the reviewer has recognized for PROSPECT and PCL analysis. However, in these cases, PCL analysis still is able to successfully predict MOA, agnostic to the peculiarities of these cases because it is reference-based. In the first instance, there are compounds that can interfere with the protein degradation knockdown system, such as ClpXP inhibitors or tetracyclines (which are part of the *sspB* induction system for degradation). In this case, reference-based predictions can still accurately predict MOA of these cases because they will still cluster with known reference compounds that are ClpXP or tetracyclines, respectively.

In the second instance of prodrugs, since we only included hypomorphs of essential targets and not the pro-drug activating enzymes, PCL analysis also works by clustering compounds with inhibitors in the reference set sharing the same end-direct target. For example, thioacetazone and isoniazid cluster with HadABC and InhA inhibitors, respectively. In fact, protonamide clusters with isoniazid, both being InhA inhibitors despite their having different activating mechanisms.

3. Along similar lines, can the authors speculate about the antitubercular leads with known MOA that they were not able to predict. Are these compounds clustered around specific targets and/or are there reasons why their MOA might be more complex or different from their 'known' MOA? The reference based approach depends on how well curated the existing 'known' compounds are and the PCL analysis is a great tool to fact check the reference compounds. In addition, it might also be possible to learn whether the success rate is higher for compounds with similar scaffolds to the reference compounds.

Multiple factors can contribute to discordance between PCL analysis predictions and published MOA annotations, as discussed in lines 488-507. In our experience, the inadequacy or shortcomings of the reference set is largely responsible in many cases. When only low numbers of compounds or scaffolds are available for a particular MOA in the reference set, assigning new compounds to this MOA can be challenging. For example, among the GSK set, we were unable to accurately predict a MenG inhibitor because a molecule with that precise MOA was absent from the reference set (the reference set contained one menaquinone biosynthesis inhibitor Ro 48-8071 but that specifically inhibits MenA).

The only MOA among the annotated GSK compounds where PCL analysis did not perform well, despite good representation in the reference set, is MmpL3, where precision was 43%. In fact, PCL analysis predicted the GSK MmpL3 inhibitors as potentially targeting Alr. This discordant prediction may be because there is a true biological interaction between MmpL3 and Alr that has not yet been recognized or inhibitors of Alr in the reference set could be misannotated (their annotation is based on *in vitro*, enzymatic activity rather than true whole cell MOA evidence). Thus, the quality of the reference set annotation can pose challenges particularly when annotations in the reference set are weak at best, incorrect at worst (see reply to Reviewer 1, point 1). We have added several lines (323-327) to further clarify these issues.

In terms of structural similarity, success rates are generally high when predicting MOA for structural analogs of compounds in the reference set. Importantly, structural similarity, however, is not required as

PCL analysis is agnostic to chemical structure (see discussion of GSK QcrB inhibitors, lines 328-336, **Extended Data Table 2**, and lines 531-551 in the discussion).

4. In the original manuscript the authors state that hypomorphs were generated either by conditional proteolysis (targeting proteins to ClpXP) or transcriptional control. What fraction of hypomorphs fall into each method of control? Is this a source of variability in the ease or difficulty of predicting MOA?

Only 2 hypomorphs under transcriptional control contributed to the current analysis (are in the 340 strains that were present in and passed strain QC in all screening waves). 11 other transcriptionally controlled hypomorphs are in the strain table (**Supplementary Data 2**) but were not included in the strain pool in all screens or grew slowly in a screening wave and were thus excluded from analysis. The remainder of the strains were all created using conditional proteolysis. We do not think this is a source of variability in predicting MOA, given their small proportional representation and that PCL prediction does not rely on the behavior of any single strain.

5. The spectral clustering and PCL analysis is conducted using dose response curves. Is it possible to comment on how this increases the accuracy of the method? i.e. is the expectation that the MOA is concentration-dependent? Using only a single dose would reduce the dimensionality and complexity of the data and facilitate the analysis of other variables.

To clarify, each data point used in spectral clustering and PCL analysis is in fact a single compound-dose combination. The purpose of screening in dose response is to ensure that all compounds are captured across one (or ideally several) doses that optimally reveal differences in strain-specific activities. Too high a dose and all strains are inhibited, and too low a dose may not reveal any sensitization.

Reducing the dimensionality could be achieved by applying objective criteria to choose the “best dose”, but this would undercut an important strength of our method: the corroboration we see when multiple adjacent doses of a compound produce similar CGI profiles that cluster together.

6. It looked like the Hill slope from the concentration response plots was allowed to float(?) If so does the Hill slope provide any additional insight into MOA and is it useful for clustering? Presumably compounds with the same MOA should have similar Hill slopes.

As mentioned in the response to 5, PCL analysis successfully clusters CGI profiles at the level of single compound-dose combinations. The reviewer is absolutely correct that quite a bit of information is captured in complete dose response curves beyond simply MIC shifts, including changes in hill slope. We are currently developing methods for reference-free prediction that integrate the information captured across the full dose-response curve.

7. Only a single growth condition was used. This is clearly an important variable and it is obviously impossible to screen all compounds under all growth conditions known to affect target vulnerability. However, is it possible to speculate how the choice of growth condition might bias the results? For instance, could this partly account for the preponderance of QcrB actives?

The reviewer raises an excellent point regarding growth conditions. For example, Kalia *et al.* (Kalia *et al.* 2019. *Sci Rep* 9: 8608) have shown that the presence of glycerol reduces efficacy of QcrB inhibitors. Thus, the conditions used in the GSK screen (glucose as carbon source) to discover their set of active compounds and in our PROSPECT screen (acetate as carbon source), may indeed have increased the relative efficacy of QcrB inhibitors. However, the differences in MIC are not so markedly different (shifts of 2-3-fold) so are unlikely to have significantly impacted the screening and discovery of compounds by GSK. We now have added text addressing this concept on lines 568-576.

8. Additionally, it would be great if the authors were able to comment on the singletons. It seems that there are singletons in the dataset that were derived from good quality measurements but weren't included due to the long distance from other compounds in the interaction map. Nonetheless, could these singletons represent completely novel MOAs? Maybe the authors could comment.

Singletons (CGI profiles that do not cluster with any reference set CGI profiles) are found among both the reference compound set and the test set. When they appear in the reference set, they are not used as anchors for MOA prediction since they lack corroboration.

When singletons appear in the test set, as long as the data is of good quality (*i.e.*, replicate reproducibility) with an informative chemical genetic interaction profile, they may indeed be reporting on a novel MOA. We tried to suggest that this is another important feature of PCL analysis of PROSPECT data, that it can quickly call out molecules with potentially new MOAs (lines 553-555, 606-615).

9. Three datasets of compounds were used - 437 from data mining, 173 from GSK, and >5000 compounds from the original PROSPECT paper. It would be nice to have a schematic cartoon clarifying all the connections between three datasets and outcomes of the analyses.

We thank the reviewer for this suggestion. The relationships are: all 437 compounds were used for the training set, while we believe that the outcomes of the analysis of the 173 GSK compounds are shown in **Fig. 3a**. Meanwhile, given that we do not extensively discuss the entire outcome of the 5000-compound set, we have chosen not to include such a schematic cartoon. However, if the reviewer feels strongly, we can reconsider.

10. Historically HTS of compound series have also revealed multiple hits for MmpL3 and DprE1 however QcrB is heavily represented in the output. Could the authors speculate/discuss why?

This is a great question and gets at a more fundamental question of why reported compounds seem to hit MmpL3, DprE1, and QcrB again and again. It could be that the frequent bias for MmpL3 and DprE1 in the literature is because of the ease of generating resistant mutants and thus elucidating MOA as a prerequisite for acceptance for publication, without a true enrichment in unbiased screening which is why we don't see it enriched here in the GSK set. In contrast, we might speculate that QcrB is truly hit frequently by diverse scaffolds resulting in its enrichment. We wonder if QcrB being a protein that is part of a big complex that requires exquisitely accurate mobility to transfer an electron might be more

susceptible to many small molecule “binders” that can disrupt this accurate transfer. But this is pure speculation.

11. Please check references. For instance ref 22 in the following text should be 76.

“To address this problem, we implemented a metric of (dose-dependent) compound-induced growth rate, GR, originally developed to measure the effect of a chemical perturbation on a collection of cancer cell lines exhibiting a range of proliferation rates (22).” This is not reference 22 but 76.

We apologize for the error and have corrected it.

12. Can the authors describe the ‘optimization’ of BRD4310 to BRD4310a? Improving cell-based activity is an important component of TB drug discovery. Are there any lessons to be learnt?

We have done some SAR study on BRD4310. For example, we know that the pyrrolidine moiety is critical for activity. Replacing the 2-nitrogen on the pyrazole ring with a carbon enhances potency without compromising microsomal stability, whereas modifying the essential 5-nitrogen on the pyrimidine ring abolishes activity. Para- and meta-substitutions on the aryl ring substantially improve potency, including activity against wild-type strains, while ortho-substitution and branching are detrimental. A methyl group is tolerated, and several other substitutions have no negative impact on stability. The optimized analog, 4310a, demonstrates strong potency and an overall favorable SAR profile. However, there are no generalizable lessons to be learned as these details are specific to the particular scaffold, and thus we have elected not to include them as we feel they are a distraction to the main point of the manuscript.

13. Please check the reported NMR chemical shifts for BRD4310a since a proton might be missing.

We apologize for the error and have corrected this (**Supplementary Note 14**).

14. Please give chemical formulas for BRD4310 and BRD4310a.

We have now included the chemical formulas for BRD4310 and BRD4310a in the revised **Supplementary Note 14**.

15. Iron acquisition (EntC). Isochorismate is used for both mycobactin and menaquinone biosynthesis. Is the EntC gene described here in the gene cluster for mycobactin biosynthesis (as implied by 'iron acquisition') or for menaquinone biosynthesis? This section of the manuscript also discusses MenC which is in the menaquinone biosynthesis pathway. It would be useful if the authors could add a sentence of two about this.

EntC (MenF) is encoded by Rv3215, which is located far from both the menaquinone biosynthetic cluster and the mycobactin biosynthetic cluster and has no paralogs in the H37Rv genome. Thus, its role as an isochorismate mutase is likely important for both pathways. This has now been clarified in the text (lines 371-372).

16. MIC90 is normally taken to indicate the MIC of 90% strains tested.

We agree and have revised the text to refer to MIC instead of MIC90. We have also clarified in the Methods that MIC in this study is defined as the minimum concentration causing 90% inhibition of bacterial growth (lines 752-753).

Reviewer #5:

We thank the reviewer for recognizing that “the manuscript by Bond et al presents an interesting and highly relevant concept” and that “the paper is well-written and introduces a number of innovative concepts in high-throughput chemical biology data analysis.”

Minor concerns

1. It is not clear what is the magnitude of variance in their proteolytic knock-downs, and what is the biological impact of the same.

As explained in the original manuscript first describing PROSPECT (Johnson *et al.* 2019. *Nature* 571: 72-78), both the degree of knockdown and the vulnerability of a particular target (*i.e.*, how much chemical inhibition of a particular target is needed to kill) are variables that impact strain behaviors in the screen. While we do not have the level of protein knockdown on a strain-by-strain basis, in selecting the knockdown strains to include in the PROSPECT screening pool, we evaluated as many as 6 candidates that expressed SspB to varying levels, providing a range of knockdown. We selected strains with the highest level of knockdown that still allowed for sufficient growth during the assay to enable rigorous statistics to be applied for identifying hits. While we recognize the fact that every strain/target has not been individually optimized to ensure it will be informative in the screen, we hope that it is evident here and in the previous publication that PROSPECT is able to successfully generate outputs of new scaffolds for both known and novel targets.

2. The potential impact of pleiotropic effects is not considered in data analyses. For example, INH may impact more than one pathway and therefore have multiple MOAs. Additionally, the inhibition of a particular pathway may require compensatory changes at the level of the DNA, RNA or protein for Mtb to survive, which could potentially be missed.

The reviewer is correct that pleiotropic effects can complicate CGI profiles. For “dirty” compounds with multiple simultaneous MOAs, PCL analysis is unlikely to be able to assign an MOA with high confidence using the reference database, unless another compound in the database shares the same “dirty” MOA profile.